# SOX-5 activates a novel RORγt enhancer to facilitate experimental autoimmune encephalomyelitis by promoting Th17 cell differentiation

Yi Tian [1,8], Chao Han[1,8], Zhiyuan Wei[1,2], Hui Dong[1], Xiaohe Shen[3], Yiqiang Cui[4], Xiaolan Fu[1], Zhiqiang Tian[1], Shufeng Wang[1], Jian Zhou[1], Di Yang[1], Yi Sun[2], Jizhao Yuan[1], Bing Ni [5,6,7] & Yuzhang Wu [1]

T helper type 17 (Th17) cells have important functions in the pathogenesis of inflammatory and autoimmune diseases. Retinoid-related orphan receptor-γt (RORγt) is necessary for Th17 cell differentiation and functions. However, the transcriptional regulation of *RORγt* expression, especially at the enhancer level, is still poorly understood. Here we identify a novel enhancer of *RORγt* gene in Th17 cells, RORCE2. RORCE2 deficiency suppresses RORγt expression and Th17 differentiation, leading to reduced severity of experimental autoimmune encephalomyelitis. Mechanistically, RORCE2 is looped to *RORγt* promoter through SRY-box transcription factor 5 (SOX-5) in Th17 cells, and the loss of SOX-5 binding site in RORCE abolishes RORCE2 function and affects the binding of signal transducer and activator of transcription 3 (STAT3) to the *RORγt* locus. Taken together, our data highlight a molecular mechanism for the regulation of Th17 differentiation and functions, which may represent a new intervening clue for Th17-related diseases.

[1] Institute of Immunology, PLA, Third Military Medical University (Army Medical University), 400038 Chongqing, People's Republic of China. [2] The First Affiliated Hospital of Third Military Medical University (Army Medical University), 400038 Chongqing, People's Republic of China. [3] Department of Microbiology and Immunology, Shanxi Medical University, 030001 Taiyuan, People's Republic of China. [4] Department of Histology and Embryology, Nanjing Medical University, Nanjing 211166 Jiangsu, People's Republic of China. [5] Department of Pathophysiology, Third Military Medical University (Army Medical University), 400038 Chongqing, People's Republic of China. [6] Key Laboratory of Extreme Environmental Medicine, Ministry of Education of China, 400038 Chongqing, People's Republic of China. [7] Key Laboratory of High Altitude Medicine, PLA, 400038 Chongqing, People's Republic of China. [8] These authors contributed equally: Yi Tian, Chao Han. ✉email: nibing@tmmu.edu.cn; wuyuzhang@tmmu.edu.cn

t is well known that T helper type 17 (Th17) cells are a subset of the T helper (Th) lineage characterized by the production of interleukin (IL)-17A. Th17 cells are crucial in host defense against extracellular pathogens and the maintenance of homeostasis in mucosal tissue[1–4]. In addition to IL-17A, IL-17F, IL-21, IL-22, and other chemokines, which enhance the migration of neutrophils to sites of infection and mediate the elimination of pathogens, can also be secreted by Th17 cells[5,6]. Although Th17 cells play an important role in host defense, they are also associated with autoimmune and inflammatory diseases, such as multiple sclerosis (MS), rheumatoid arthritis (RA), inflammatory bowel disease (IBD), systemic lupus erythematosus (SLE), psoriasis, allergy, and asthma[7,8].

The differentiation of naïve CD4$^+$ T cells into Th17 cells is mainly regulated by IL-6 and transforming growth factor beta (TGF-β)[9,10]. The binding of IL-6 to its cognate receptor on the Th17 cell membrane results in signal transducer and activator of transcription 3 (STAT3) phosphorylation and dimerization[3]. The phosphorylated STAT3 dimer subsequently translocates into the nucleus and induces the expression of the transcription factor (TF) retinoid-related orphan receptor-γt (RORγt), which is encoded by the *RORC* gene[11,12]. RORγt is considered a lineage-specific marker of Th17 cells and is required for Th17-cell lineage commitment and function[11]. RORγt can directly activate the genes encoding IL-17A and IL-17F[11]. RORγt-deficient (RORγt$^{-/-}$) mice have reduced Th17 differentiation and lowered susceptibly to experimental autoimmune encephalomyelitis (EAE)[11]. Due to the critical role of RORγt in Th17 cells, it is crucial to better understand the transcriptional and posttranscriptional regulation of RORγt gene expression[13]. Enhancers are one of the critical cis-regulatory elements at the transcriptional level[14]. Exploring the potential enhancers involved in regulating RORγt gene expression in Th17 cells is a pivotal issue for understanding Th17-related diseases.

The identification of potential enhancers is facilitated by specific histone modifications that mark enhancer-like regions in the genome. In particular, active enhancers display overenrichment of histone H3 lysine 4 monomethylation (H3K4me1), H3K4 dimethylation (H3K4me2), and H3 lysine 27 acetylation (H3K27ac) but not H3K4 trimethylation (H3K4me3)[15,16]. Enhancers usually regulate the transcriptional activity of a given gene regardless of their position, orientation, or distance from the target promoter. This regulation generally occurs through a chromatin-looping mechanism through which the enhancer comes into close proximity with the target promoter[17]. The formation of a chromatin loop relies on the binding of multiple tiers of TFs to enhancers, which leads to the opening of the chromatin and recruitment of Mediator (Med), an important protein complex that mediates the interaction between TFs and RNA polymerase II (RNA pol II)[14,17–19].

In the present study, we identify RORCE2 as a novel enhancer of the RORγt gene in mouse Th17 cells. SRY-box transcription factor 5 (SOX-5) mediates the looping between RORCE2 and the RORγt gene promoter to promote Th17 differentiation and EAE pathogenesis. Furthermore, SOX-5 is a prerequisite for STAT3 binding to RORCE2 and exerting its TF function. The present study will provide not only a new mechanism underlying Th17 differentiation but also a potential clue for intervention in Th17-related diseases in the future.

## Results

**Identification and characterization of a novel active enhancer of the *RORγt* gene.** Since H3K4me2 is a well-known histone marker for active enhancers[16], we took advantage of H3K4me2 chromatin immunoprecipitation sequencing (ChIP-seq) data for mouse Th17 cells and non-Th17 cells including Th1 and innate

lymphoid cells (ILCs) in the GEO database[20,21] to identify potential enhancers of the *RORγt* gene. Intriguingly, we found a unique H3K4me2 peak associated with the *RORγt* locus specifically in Th17 cells but not in Th1 cells or ILCs. The sequence element, named RORCE, was located between −7 kb and −3.6 kb upstream of the *Rorc* gene and might serve as a potential enhancer in Th17 cells (Fig. 1a). By comparing the chromatin signatures of the syntenic region of mouse RORCE between human Th17 and Th0 cells, we found that active enhancer-associated epigenetic markers, including H3K4me1 and H3K27ac, were also enriched in human Th17 cells (Supplementary Fig. 1) (http://egg2.wustl.edu/roadmap/web_portal/)[22]. Thus, these ChIP-seq data suggested that RORCE might be an active RORγt enhancer in Th17 cells.

Considering the median size of common enhancers is 1.3 kb[23], we subdivided RORCE into three consecutive fragments, i.e., RORCE1 (−7 kb to −5.8 kb), RORCE2 (−5.8 kb to −4.6 kb), and RORCE3 (−4.6 kb to −3.6 kb) (Supplementary Fig. 2a). To evaluate our prediction, we used a ChIP coupled with quantitative PCR (ChIP-qPCR) assay to examine the profiles of other epigenetic markers of an active enhancer including H3K4me1, H3K27Ac, and H3Ac in RORCE1, RORCE2, and RORCE3 with fluorescence activated cell sorting (FACS)-purified mouse Th17 and non-Th17 cells (Supplementary Figs. 2b and 3a). Our results showed that among RORCE1, RORCE2, and RORCE3, these enhancer-associated epigenetic markers were more significantly enriched in RORCE2 of the Th17 cells than in that of the non-Th17 cells (Supplementary Fig. 2c–e), suggesting that RORCE2 might be an active enhancer in Th17 cells.

We next determined whether RORCE2 acts on the *RORγt* promoter using a dual-luciferase assay in EL4 murine tumor T cell line which constitutively expresses RORγt and IL-17A under resting conditions[24] and human embryonic kidney (HEK) 293T cell line[25,26]. The six potential regulatory elements, including RORCE1, RORCE2, RORCE3, and their combinations, were cloned upstream of the *RORγt* promoter in pGL3 vectors (Fig. 1b)[27]. The results showed that construct only containing RORCE2, compared with other constructs, could enhance the transcriptional activity of the *RORγt* promoter by 2–3 folds in both EL4 and 293T cells (Fig. 1c and d). We further detected the luciferase activity of these constructs in Th17-polarized cells. Results showed that all the constructs containing RORCE2 were capable of enhancing the luciferase activity (Fig. 1e). When these six fragments were cloned downstream of the luciferase gene in the reporter constructs, they also showed a similar activity in Th17-polarized cells (Fig. 1f). These results indicated that RORCE2 might act as a candidate enhancer for *RORγt* gene in a position-independent manner. All of these in vitro results suggested that RORCE2 was a potent enhancer element for the *RORγt* promoter in Th17 cells that might contribute to Th17 differentiation and functions.

**RORCE2 is required for proper Th17 cell differentiation through upregulating *RORγt* gene expression.** To investigate the effect of RORCE2 on the regulation of *RORγt* gene expression and Th17 differentiation in vivo, we generated RORCE2-deficient (RORCE2$^{-/-}$) mice by CRISPR/Cas9 genome editing (C57BL/6 line, Beijing Biocytogen Co., Ltd.) (Fig. 2a). To exclude the possibility of off target caused by CRISPR-Cas9, we detected the top 10 of potential off-targets of upstream/downstream sgRNAs in RORCE2$^{-/-}$ mice and found that there were not any deletions or mutations (Supplementary Fig. 4 and Table 1). We found that the mRNA and protein expression levels of RORγt, which were measured by reverse transcription followed by qPCR (RT-qPCR) in splenic CD4$^+$ T cells and evaluation of the mean fluorescence

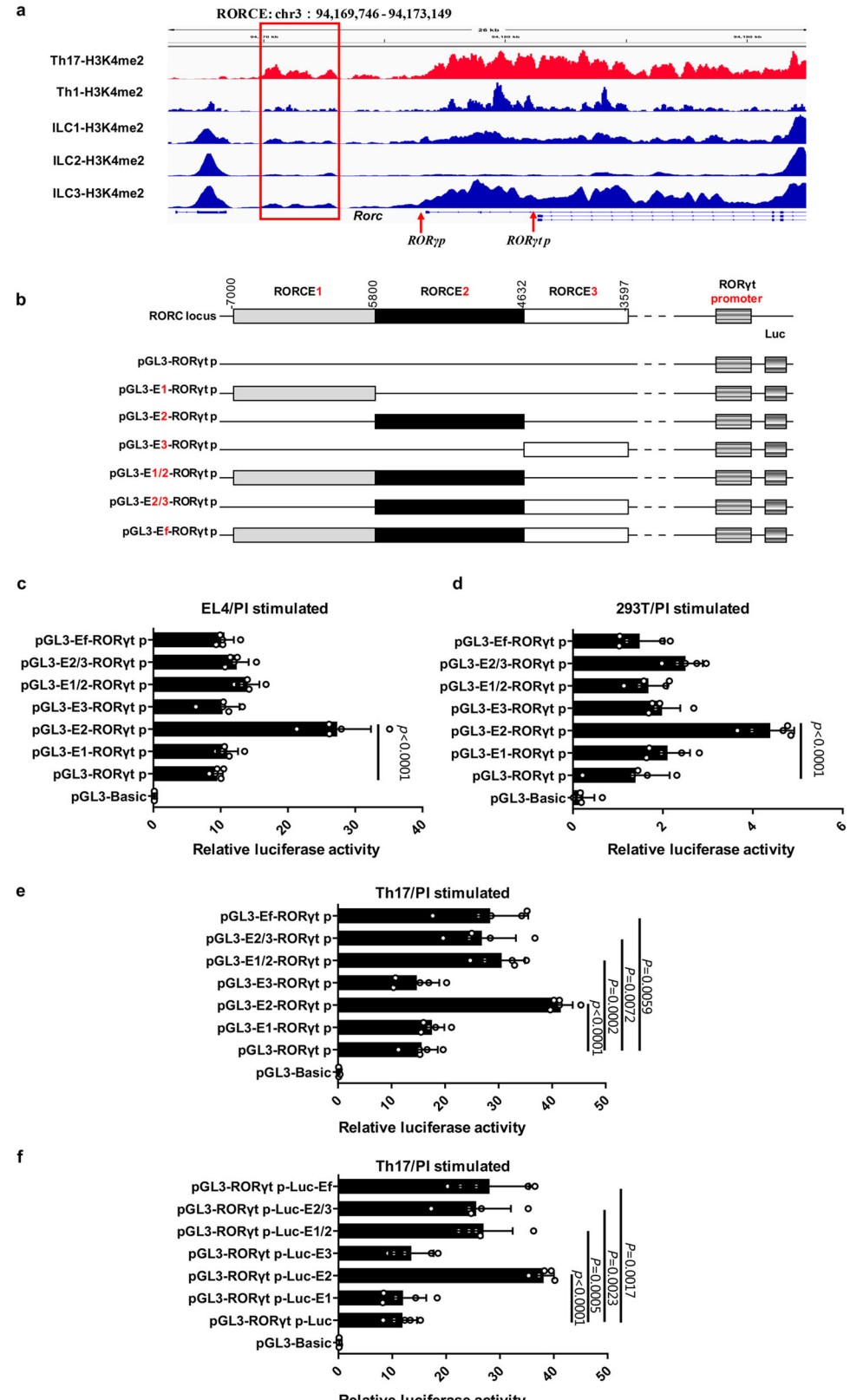

intensity (MFI) in splenic CD4$^+$ROR$\gamma$t$^+$Th17 cells, were decreased by about 75% and 50% in RORCE2$^{-/-}$ mice compared with wild-type (WT) mice, respectively (Fig. 2b–d and Supplementary Fig. 3b). Accompanying the decrease in ROR$\gamma$t expression, the frequency of CD4$^+$ROR$\gamma$t$^+$ Th17 cells was decreased by about 50% in the splenic CD4$^+$ T cells of RORCE2$^{-/-}$ mice

(Fig. 2c and d). Because ROR$\gamma$t is required for the induction of IL-17A transcription in Th17 cells[11], we found that RORCE2 deficiency resulted in significant decreases in the expression of IL-17A and frequency of CD4$^+$IL-17A$^+$ Th17 cells in the splenic CD4$^+$ T cells (Fig. 2e–g and Supplementary Fig. 3b). Similar results were observed in the lamina propria (LP) in mouse small

**Fig. 1 Identification and characterization of a novel active enhancer of the *RORγt* gene. a** IGV browser views of H3K4me2 profiles of RORCE (chr3: 94,169,746-94,173,149), which is indicated by red box, in Th17 cells, Th1 cells, and ILCs. RORγt and its isoform RORγ are encoded by *Rorc* locus through the activation of alternative promoters *RORγt* promoter (*RORγt p*) and *RORγ* promoter (*RORγ p*), respectively. **b** RORCE1 (chr3: 94,169,746-94,170,945), RORCE2 (chr3: 94,170,946-94,172,113), RORCE3 (chr3: 94,172,215-94,173,149), RORCE1/2 (chr3: 94,169,746-94,172,113), RORCE2/3 (chr3: 94,170,946-94,173,149), and full-length RORCE (RORCEf, chr3: 94,169,746-94,173,149) were cloned upstream of *RORγt* promoter (*RORγt p*) in pGL3 vectors. **c-e** Dual-luciferase assays of the above six reporter constructs (**b**) in EL4 (**c**), 293T cells (**d**), or Th17-polarized cells (**e**). 293T, EL4, and Th17-polarized cells were transfected with indicated plasmids and cultured at 37 °C. After 24 h of transfection, cells were stimulated with PI for 4–5 h. The cells were harvested, lysed, and assessed with dual-luciferase reporter system. Renilla luciferase activity of pRL-TK was used for normalization. **f** RORCE1, RORCE2, RORCE3, RORCE1/2, RORCE2/3, and RORCEf were cloned downstream of the luciferase gene in pGL3-*RORγt p* vectors, and then Dual-luciferase assays of various reporter constructs in Th17-polarized cells were performed. Mean ± SEM are shown, $n = 5$ independent experiments, unpaired two-tailed Student's *t*-test (**c-f**). Experimental mice were between 8 and 12 weeks of age, with no preference to gender and were maintained on a C57BL/6 background. Source data are provided as a Source Data file.

intestine (Fig. 2h–n and Supplementary Fig. 3c). However, we did not find any changes in the mRNA expression of the Th1 signature gene *T-box transcription factor 21* (*Tbx21*) or the Th2 signature gene *GATA binding protein 3* (*Gata3*) after RORCE2 knockout or in the frequencies of Th1 and Th2 cells of splenic CD4$^+$ T cells (Supplementary Figs. 5 and 3b). In addition, considering the similarities between the ILC3 and Th17 cells, such as the specific expression and critical role of RORγt in both types of cells, we also detected the effect of RORCE2 on ILC3 cells. In contrast to the results for Th17 cells, the relative mRNA expression level of *RORγt* and ILC3 cell frequency were unchanged in the LP of RORCE2$^{-/-}$ mice compared with that of WT mice (Fig. 2o–q and Supplementary Fig. 3d). These data suggested that RORCE2 might favor T cell differentiation toward the Th17 cell lineage in vivo.

To further elucidate the contribution of RORCE2 to Th17 cell differentiation, we polarized naïve RORCE2$^{-/-}$ or WT CD4$^+$ T cells into Th17 cells. The frequency of CD4$^+$RORγt$^+$ Th17 cells or CD4$^+$IL-17A$^+$ Th17 cells generated from RORCE2$^{-/-}$ T cells and the MFI of RORγt in CD4$^+$RORγt$^+$Th17 cells were significantly lower than the values of the same parameters for WT T cells (Fig. 3a–e and Supplementary Fig. 3e), which was in accordance with the reduced IL-17A production in cultured RORCE2$^{-/-}$ T cells (Fig. 3f). In addition, the mRNA expression of *RORγt* or *IL-17A* was significantly downregulated in RORCE2$^{-/-}$ T cells under Th17-polarizing conditions (Fig. 3g–h). However, RORCE2 deficiency did not affect Th1 or Th2 cell differentiation in vitro, including cell frequency, mRNA expression of key signature genes (*Tbx21* in Th1 cells or *Gata3* in Th2 cells), and cytokine secretion (interferon gamma (IFNγ) in Th1 cells or IL-4 in Th2 cells) (Fig. 3i–p and Supplementary Fig. 3e) under the respective polarization conditions. In summary, these results indicated that RORCE2 played a positive role in Th17 cell differentiation in vitro by regulating the expression of the *RORγt* gene.

**RORCE2 deficiency leads to decreased EAE severity.** We next determined whether RORCE2 deficiency impacts the development of EAE, a Th17-dependent inflammatory disease, by immunizing 10- to 12-week-old RORCE2$^{-/-}$ and WT mice with the myelin oligodendrocyte glycoprotein 33–35 (MOG 33–35) peptide (emulsified in complete Freund's adjuvant (CFA)) and pertussis toxin[28]. The disease progression in the RORCE2-deficient mice was remarkably less than that in their WT littermates (Fig. 4a). Histological analysis revealed that the degrees of inflammatory cell infiltration and demyelization in the spinal cord were significantly alleviated in the RORCE2-deficient mice (Fig. 4b). The reduction in cell infiltration was further confirmed by flow cytometry analysis of T cells isolated from the spinal cord of the diseased mice (Fig. 4c–e and Supplementary Fig. 3f). The absolute numbers of total CD4$^+$ T cells and CD4$^+$IL-17A$^+$ Th17

cells and the frequency of CD4$^+$IL-17A$^+$Th17 cells were markedly reduced in the spinal cords from the RORCE2-deficient mice compared with those from the WT mice (Fig. 4c), reflecting the reduced disease severity in the RORCE2-deficient mice. However, the percentages of CD4$^+$IFNγ$^+$ Th1 and CD4$^+$IL-4$^+$ Th2 cells in the spinal cords from the RORCE2$^{-/-}$ mice were comparable to those in the spinal cords from the WT mice (Fig. 4d and e). To further confirm the decreased disease severity of RORCE2$^{-/-}$ mice was Th17-related, we immunized RORCE2$^{-/-}$ and WT mice with MOG 35–55 peptide and extracted the draining lymph nodes on day 8. Isolated cells from the lymph nodes were further cultured in ex vivo with MOG for 3 days. The concentrations of IL-17A, IFNγ, and IL-4 were measured by enzyme-linked immunosorbent assay (ELISA), respectively. We found that IL-17A production was decreased in RORCE2$^{-/-}$ mice, whereas IFNγ and IL-4 production was unaffected (Fig. 4f). These data further support the significance of RORCE2 in the generation of Th17 cells and the production of IL-17.

**SOX-5 is required for the looping between RORCE2 and the *RORγt* promoter in Th17 cells.** Since it is well known that the regulatory activity of an enhancer mainly depends on the formation of a chromatin loop between the enhancer and its target promoter[17], we used a chromosome conformation capture-qPCR (3C-qPCR) assay[29] to determine whether RORCE2 interacts with the *RORγt* promoter in Th17 cells. Restriction enzyme mapping analysis of the RORCE region and a literature search[29] led us to select the NlaIII restriction enzyme to conduct the 3C-qPCR assay, allowing for high-resolution analysis of this short sequence region. Moreover, we employed the single NlaIII site within the *RORγt* promoter as the anchor site (AS) to test the spatial proximities with other NlaIII sites scattered across the RORCE region (test sites (TSs) 1 to 8 are indicated in Fig. 5a). The relative interaction frequencies between the AS and the TS sites were then quantified by qPCR. We chose the murine T cell line EL4 for the 3C-qPCR assay because RORγt expression is constitutive under resting conditions and upregulated upon activation (Supplementary Fig. 6a and b). B16, a melanoma tumor cell line without RORγt expression, was used as a negative control (Supplementary Fig. 6a and b). The results showed that the AS interacted most frequently with TS3 located in the RORCE2 region among all TS sites tested in both PMA/ionomycin (PI)-simulated EL4 cells and unstimulated EL4 cells (Fig. 5b). However, there was no interaction between the AS and the TS sites in B16 cells (Fig. 5b). Comparable results were observed in similar comparisons (FACS-purified Th17 cells vs. non-Th17 cells or Th17-polarized cells vs. naïve cells; Fig. 5c and d). In addition, we did not observe any loop formation in Th1- or Th2-polarized or naïve T cells derived from WT mice (Fig. 5e and f). These results strongly suggested that the RORCE2 region was in close proximity to the *RORγt* gene promoter in Th17 cells.

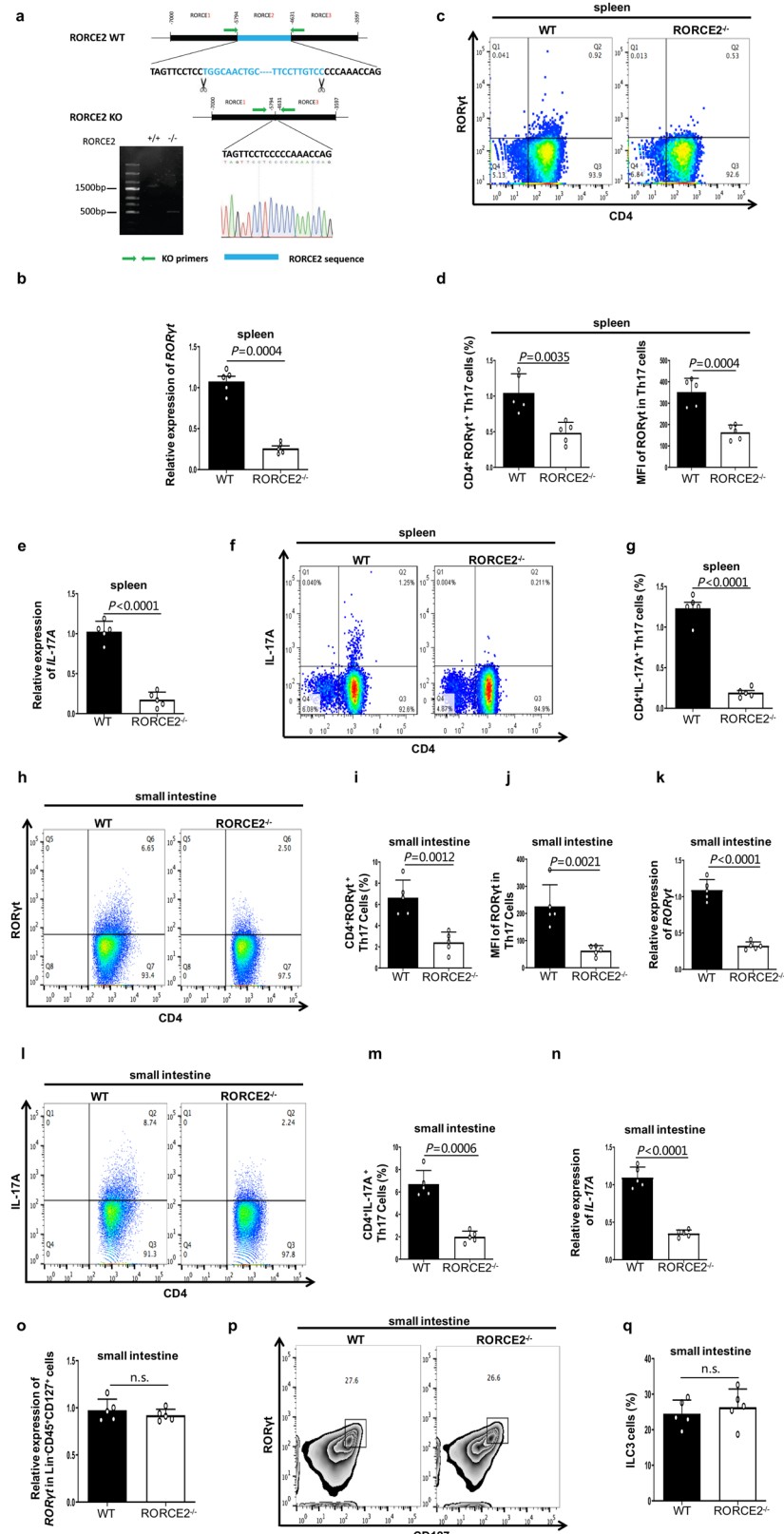

Since TFs are required for the formation of a chromatin loop between an enhancer and its target promoter[17], we next investigated which TF(s) was involved in the interaction between TS3 in RORCE2 and the AS in the *RORγt* gene promoter in Th17 cells. To this end, the JASPAR database (http://jaspar.genereg.net) was used to rank all TFs that might bind to RORCE (Fig. 6a). Of all

candidates, SOX-5 emerged as the top candidate due to it having the highest binding score (Fig. 6a) as well as the closest location to TS3 (Fig. 6b). Supporting this notion, SOX-5 has also been implicated in Th17 cell differentiation via binding to the *RORγt* promoter[30].

First, the expression of *Sox5* in PI-stimulated EL4 cells and Th17-polarized cells was confirmed (Supplementary Fig. 7). Using

**Fig. 2 Loss of RORCE2 significantly decreases RORγt expression and Th17 cell numbers in vivo. a** The sequences of RORCE2 deleted with CRISPR/Cas9 technology (top). Allele knockout (KO) was confirmed by PCR (bottom left) and sequencing (bottom right). **b** Relative mRNA expression of *RORγt* gene in splenic CD4⁺ T cells from RORCE2⁻/⁻ (KO) or wild-type (WT) mice were measured by RT-qPCR assay. **c, d** Cells sorted from spleens of indicated mice were stained for CD4 and RORγt. The frequencies of CD4⁺RORγt⁺ Th17 cells in RORCE2⁻/⁻ and WT mice and mean fluorescence intensity (MFI) of RORγt in Th17 cells were analyzed by flow cytometry (**c**) and statistically evaluated (**d**). The numbers in quadrants indicate the percentage of cells in the quadrant. **e** Relative mRNA expression of *IL-17A* gene in splenic CD4⁺ T cells from indicated mice. **f, g** Frequencies of CD4⁺IL-17A⁺ Th17 cells in splenic CD4⁺ T cells from indicated mice. **h, i** RORγt⁺ Th17 frequencies in CD4⁺CD45⁺ Lin⁺ lymphocyte population from LP of small intestine of indicated mice. **j, k** MFI of RORγt in LP Th17 cells (**j**) and relative mRNA expressions of *RORγt* gene (**k**) in LP CD4⁺ T cells from the small intestine of indicated mice. **l, m** IL-17A⁺ Th17 frequencies in the CD4⁺CD45⁺ Lin⁺ lymphocyte population of LP from the small intestine of indicated mice. **n** Relative mRNA expressions of *IL-17A* gene in LP CD4⁺ T cells from small intestine of indicated mice. **o–q** Relative expression of *RORγt* gene in CD45⁺ Lin⁻CD127⁺ lymphocyte population (**o**) and frequencies of CD127⁺ RORγt⁺ ILC3 cells in the CD45⁺Lin⁻ lymphocyte population of the lamina propria from small intestine of indicated mice (**p** and **q**). Mean ± SEM are shown, *n* = 5 independent experiments, unpaired two-tailed Student's *t*-test (**b, d, e, g, i–k, m–o, q**). Experimental mice were between 8 and 12 weeks of age, with no preference to gender and were maintained on a C57BL/6 background. Source data are provided as a Source Data file.

ChIP-qPCR, we found that SOX-5 bound to both RORCE and the *RORγt* promoter in PI-stimulated EL4 cells or WT Th17-polarized cells compared to B16 cells or naïve CD4⁺ T cells, respectively (Fig. 6c and d). Furthermore, a ChIP-3C assay (also called a ChIP-loop assay) demonstrated that SOX-5 was involved in the process of chromatin loop formation between TS3 and the *RORγt* promoter in both PI-stimulated EL4 cells and Th17-polarized cells (Fig. 6e and f). This SOX-5-mediated loop did not exist in Th1- or Th2-polarized cells from either WT (Fig. 6g and h).

To further characterize the role of SOX-5 in the chromatin loop in vivo, we generated SOX-5-BS-deficient (SOX-5-BS⁻/⁻) mice by CRISPR/Cas9-mediated genome editing (C57BL/6 line, Fig. 6i). We found that SOX-5-BS deficiency in RORCE resulted in a significant decrease in SOX-5 enrichment (Fig. 6j). Intriguingly, 3C-qPCR and ChIP-loop assays showed that the degree of TS3-*RORγt* promoter interaction was also significantly reduced by deleting the SOX-5-BS in the promoter region (Fig. 6k and l), confirming that the interaction between RORCE2 and the promoter of the *RORγt* gene was mainly mediated by SOX-5.

**SOX-5-BS deletion significantly reduces the differentiation of Th17 cells.** To better understand the effect of the SOX-5-mediated loop between TS3 and the *RORγt* promoter on the regulation of RORγt expression and Th17 differentiation, we further investigated the effect of SOX-5-BS ablation on Th17 cells. The overall phenotype largely resembled that of RORCE2 deficiency. In the spleen, both the mRNA and protein levels of RORγt and IL-17A were decreased by about 60% in SOX-5-BS-deficient mice compared to WT mice in vivo (Fig. 7a–c and Supplementary Fig. 3b). The percentages of Th17 cells stained with anti-CD4 and anti-RORγt or anti-IL-17A mouse antibodies were also significantly reduced in the splenic CD4⁺ T cells (Fig. 7b and d and Supplementary Fig. 3b). Similar results were observed in the LP of mouse small intestine (Fig. 7e–h and Supplementary Fig. 3c). SOX-5-BS deficiency resulted in a significantly decreased frequency of CD4⁺RORγt⁺ Th17 or CD4⁺IL-17A⁺ Th17 cells under the Th17-polarizing conditions (Fig. 8a–c and Supplementary Fig. 3e). Moreover, compared with WT Th17 cells, Th17 cells with the SOX-5-BS knocked out exhibited decreased mRNA and protein expression of RORγt and IL-17A (Fig. 8a, d, and e) and decreased secretion of IL-17A (Fig. 8f). *Sox5* overexpression caused marked increase of RORγt and IL-17A expressions in WT Th17-polarized cells (Fig. 8g). In addition, we also observed a mild increase of RORγt and IL-17A expressions in Th17-polarized cells from RORCE2⁻/⁻ and SOX-5-BS⁻/⁻ mice after *Sox5* overexpression (Fig. 8g), suggesting that SOX-5 may have other target sites in Th17 cells[30].

**SOX-5-BS deficiency alleviates the severity of EAE.** We further investigated whether SOX-5-BS deficiency affects EAE severity.

The results indicated that SOX-5-BS deficiency phenocopied RORCE2 deficiency, including significant decreases in disease progression, cell infiltration, and spinal cord demyelization (Fig. 9a and b). The frequency of CD4⁺IL-17A⁺ Th17 cells and the absolute numbers of total CD4⁺ T cells and CD4⁺IL-17A⁺ Th17 cells were considerably reduced in the spinal cord of SOX-5-BS-deficient mice compared to the spinal cord of WT mice (Fig. 9c–e and Supplementary Fig. 3f). These data strongly indicated the alleviated EAE severity in SOX-5-BS-deficient mice.

**SOX-5 and STAT3 synergistically activate RORCE2 in Th17 cells.** Since SOX-5 does not contain a transactivation domain, its activity is likely mediated by other molecules[30]. Interestingly, we found a putative DNA binding site for STAT3 (STAT3-BS), a well-known transcriptional activator, close to the SOX-5-BS (Fig. 10a). Due to the critical role of STAT3 in the induction of RORγt expression and Th17 differentiation[31–33], we investigated the contribution of STAT3 to RORCE2 enhancer function. ChIP-qPCR results showed that STAT3 more strongly interacted with RORCE2 in PI-stimulated EL4 cells and Th17-polarized cells than in B16 cells and naïve CD4⁺ T cells, respectively (Fig. 10b and c). Moreover, *Stat3* overexpression led to the increased STAT3–RORCE2 interaction and upregulation of RORγt and IL-17A expressions in WT Th17-polarized cells as evidence by RT-qPCR and/or ELISA (Fig. 10d); accordingly, *Stat3* knockdown caused the reverse results (Fig. 10d). However, this binding was significantly disrupted upon SOX-5-BS deficiency in Th17-polarized cells, suggesting that SOX-5 was required for the binding of STAT3 to RORCE2 (Fig. 10c). When RORCE2 lacked the STAT3-BS upstream of the *RORγt* promoter in the pGL3 vector, the transcriptional activity of RORCE2 was lower than that of normal RORCE2 in EL4 cells (Fig. 10e). Furthermore, co-immunoprecipitation (Co-IP) assay showed that STAT3 interacted with SOX-5 (Fig. 10f and g), suggesting that the SOX-5 might affect STAT3 recruitment to RORCE2.

**Discussion**
In this study, we identify a novel enhancer, RORCE2, for promoting *RORγt* gene expression in Th17 cells in vivo and in vitro. Further investigation shows that the interaction between RORCE2 and the *RORγt* gene promoter is mediated by SOX-5 binding. Knocking out the BS for SOX-5 in RORCE markedly decreases RORγt expression and Th17 differentiation, resulting in decreased EAE severity. Finally, we demonstrate that STAT3 cooperated with SOX-5 to participate in RORCE2-mediated gene activation.

A recent study identifies an enhancer at ~1.5 kb upstream of the *RORγt* promoter in human Th17 cells[34]. However, this study does not investigate the effect of this enhancer on Th17 differentiation or inflammatory disease in vivo. In the present study, we identified a

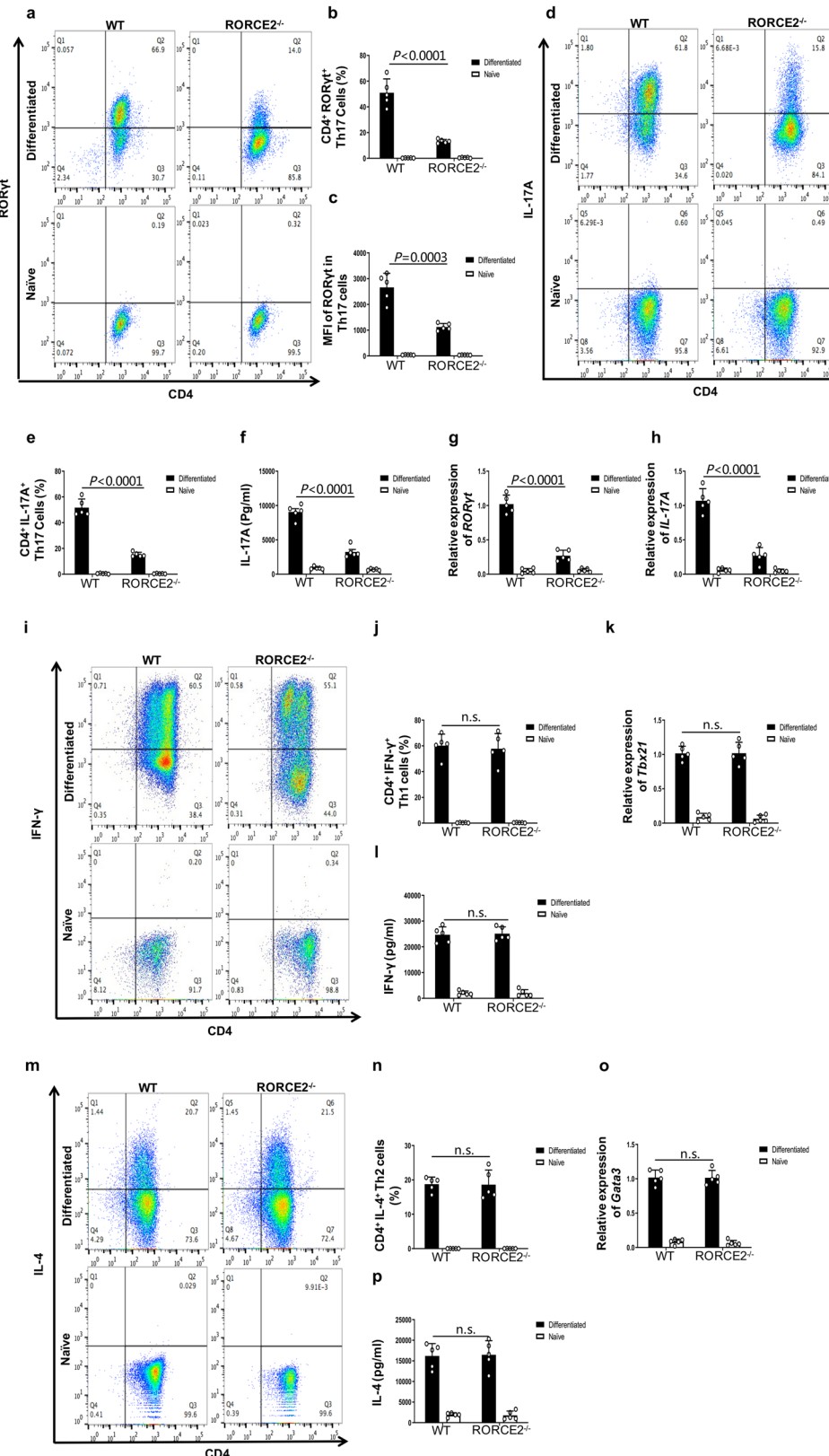

novel enhancer at 5.8–4.6 kb upstream of the *Rorc* gene in mouse Th17 cells. More importantly, we confirmed the crucial roles of this enhancer in RORγt expression, Th17 differentiation, and EAE disease promotion. Therefore, it is the first *RORγt* enhancer identified by a specific enhancer sequence-knockout approach. Furthermore, we found that RORCE2 was an active *RORγt* enhancer in Th17 cells because the Th1, Th2, and ILC3 cell linages were not affected by knocking out the RORCE2 element. Although RORγt is also the key TF of the ILC3 cell lineage, these cells exhibit much lower H3K4me2 enrichment at the RORCE2 region than Th17 cells[20]. We speculate that an enhancer other than RORCE2 is employed to promote RORγt expression in ILC3 cells.

**Fig. 3 RORCE2 deficiency inhibits Th17 cell polarization in vitro. a** Naïve splenic CD4$^+$ T cells from RORCE2$^{-/-}$ or WT mice were stimulated under Th17-polarizing conditions for 5 days. The frequency of CD4$^+$RORγt$^+$ Th17 cells was measured by flow cytometry. **b, c** Summaries for the percentages of CD4$^+$RORγt$^+$ Th17 cells (**b**) and MFI of RORγt in Th17 cells (**c**) generated from RORCE2$^{-/-}$ and WT mice on day 5 after in vitro polarization are shown. **d** Naïve splenic CD4$^+$ T cells from RORCE2$^{-/-}$ or WT mice were stimulated under Th17-polarizing conditions for 5 days. The frequency of CD4$^+$IL-17A$^+$ Th17 cells was measured by flow cytometry. **e** Summaries for the percentages of CD4$^+$IL-17A$^+$ Th17 cells generated from RORCE2$^{-/-}$ and WT mice on day 5 after in vitro polarization are shown. **f** IL-17A production in culture supernatants was measured by ELISA. **g, h** Relative expressions of *RORγt* (**g**) and *IL-17A* (**h**) genes in Th17-polarized cells were measured by RT-qPCR. **i-p** Naïve splenic CD4$^+$ T cells from RORCE2$^{-/-}$ or WT mice were polarized under Th1- or Th2-polarizing conditions, and frequencies of Th1 (**i** and **j**) and Th2 (**m** and **n**) cells were analyzed by flow cytometry on day 6 after differentiation. Relative expression of the *Tbx21* (**k**) or *Gata3* (**o**) gene in Th1- or Th2-polarized cells was analyzed by RT-qPCR. The indicated cytokine production in cell culture supernatants was measured by ELISA (**l** and **p**). Mean ± SEM are shown, $n = 5$ independent experiments, unpaired two-tailed Student's $t$-test (**b, c, e-h, j-l, n-p**). Experimental mice were between 8 and 12 weeks of age, with no preference to gender and were maintained on a C57BL/6 background. Source data are provided as a Source Data file.

It is well known that an active enhancer regulates gene expression by interacting with the corresponding target promoter[17]. In this study, we established a previously unknown interaction between RORCE2 and the *RORγt* promoter using 3C-qPCR. In addition, the looping between an active enhancer and a promoter is facilitated by TFs[17]. The binding of TFs to an enhancer is the first step for the formation of the loop, which recruits other cofactors to further mediate the interaction between promoters and enhancers[14]. Our study found an important TF (i.e., SOX-5) that participated in such an interaction with RORCE2 in Th17 cells through bioinformatic prediction and verification in multiple experiments. Previous studies investigating the roles of certain TFs in enhancer–promoter looping largely rely on knocking out or knocking down TF genes[35–38]. In contrast to the published literature, in this study, we knocked out the SOX-5-BS in RORCE by using a CRISPR-Cas9 strategy to observe the role of SOX-5 in the looping between RORCE2 and the *RORγt* promoter. This study provides direct evidence that confirms the role of TF binding in the enhancer region because knocking out a TF in a cell may result in an unexpected effect on cell biology due to the multiple targets of the TF[39].

STAT3 is crucial for Th17 differentiation[31–33]. In addition, SOX-5 itself does not contain a transactivation or transrepression domain[30,40], and thus, SOX-5 activity is likely to be mediated by the partner molecules with which SOX-5 interacts. Therefore, we investigated whether there is an association between SOX-5 and STAT3. In this study, we found a STAT3-BS closely upstream of the SOX-5-BS in RORCE2. STAT3 strongly bound to the STAT3-BS to promote *RORγt* transcription because deletion of this STAT3-BS severely downregulated transcriptional activity. Intriguingly, when the SOX-5-BS was deleted, STAT3 failed to bind to RORCE2, suggesting that SOX-5 binding to RORCE2 was a prerequisite for STAT3 binding to RORCE2 and exerting its TF function. Thus, STAT3 is probably involved in RORCE2 activation as a cofactor of SOX-5. However, how SOX-5 influences STAT3 activity needs to be further clarified in the future. In addition to SOX-5 and STAT3, other TFs might contribute to RORCE2 activity; for example, CREB1, which plays a critical role in autoimmunity by promoting Th17 differentiation[28], is predicted to be the top potential TF binding to RORCE2 by JASPAR in this study.

In addition, we observe that RORCE2 or the binding of SOX-5 to RORCE2 can not only regulate the *RORγt* gene at the transcriptional level but also impact the development of the Th17-dependent inflammatory disease EAE. This pioneering study demonstrates the key role of this enhancer in EAE, raising the significance of cis-regulatory elements in disease pathogenesis. However, there are several other Th17-related diseases, including SLE, collagen-induced arthritis (CIA), and colitis[41]. Whether RORCE2 plays a vital role in such Th17-related diseases remains to be elucidated in mouse models and human cells.

In summary, we identify RORCE2 as an enhancer of the *RORγt* gene in mouse Th17 cells in vitro *and* in vivo. SOX-5 mediates the looping between RORCE2 and the *RORγt* gene promoter to

promote Th17 differentiation and EAE pathogenesis. Furthermore, SOX-5 is a prerequisite for STAT3 binding to RORCE2 and exerting its TF function. Thus, we propose a mechanism underlying the RORCE2-mediated regulation of RORγt expression in Th17 cells by combining the findings of previous studies and our results, as shown in Supplementary Fig. 8. The signaling of the polarization cytokine IL-6 activates STAT3, which is required for the induction of SOX-5 in Th17 cells[30]. SOX-5 binds to RORCE and mediates the looping of RORCE to the *RORγt* promoter, facilitating the binding of activated STAT3 to RORCE2 to promote RORγt expression, Th17 cell differentiation, which probably occurs through a process involving other trans-acting factors, and EAE progression. The present study will provide not only a new mechanism underlying Th17 differentiation but also a potential clue for intervention in Th17-related diseases in the future.

## Methods

**Generation of RORCE2-deficient and SOX-5-BS-deficient mice**. Both RORCE2-deficient and SOX-5-BS-deficient mice were generated via CRISPR/Cas9/sgRNA-mediated gene targeting[42,43]. The intergenic sequence of RORCE2 (chr3: 94,170,953-94,172,115) was targeted by sgRNAs with the following sequences: 5′-TAGGTCTG GTTTGGGGGACA-3′ and 5′-AAACTGTCCCCCAAACCAGA-3′ for the upstream site and 5′-TAGGCCTTAGAGGCAGTTGCCAGG-3′ and 5′-AAACCCTGGCAA TGCCTCTAAGG-3′ for the downstream site. The SOX-5-BS (chr3: 94,170,884-94,170,889) was targeted by sgRNAs with the following sequences: 5′-TAGGTGTC AGCACGGAGGATTGTT-3′ and 5′-AAACAACAATCCTCCGTGCTGACA-3′. After confirming the cleavage efficiency of these sgRNAs in mouse embryonic stem (mES) cells, in vitro-transcribed mRNAs containing the sgRNA/Cas9 sequence were injected into zygotes through microinjection, followed by implantation into surrogate mice. The RORCE2-deficient model was generated by Beijing Biocytogen Co., Ltd. F0 mice were genotyped with PCR primers with the following sequences: 5′-T GAAAATCAGGAGTGGAGGGCTGGAG-3′ and 5′-CCAATTGTCCCCGTATAG GACCTGC-3′. The WT allele had a PCR product length of ~1.7 kb, and the deficient allele had a PCR product length of ~523 bp. The SOX-5-BS-deficient mice were generated by Dr. Yiqiang Cui of Nanjing Medical University. F0 mice were genotyped with PCR primers with the following sequences: 5′-TTGCTCCAGTTGT CCAC-3′ and 5′-ATCTGTCTAAGGGCGAAG-3′. These PCR products were used for sequencing. All mice were housed in groups of 4–5 per cage in standard closed plastic cages containing bedding, enrichment, food, and water, at controlled stable room temperature and humidity, light/dark cycle 12 h per day. All the experimental/control mice were maintained on a C57BL/6 background under specific pathogen-free conditions and were bred separately. Experimental mice were between 8 and 12 weeks of age with no preference to gender, and were deeply anesthetized with sodium pentobarbital (i.p. 100 mg/kg, Sanofi) and rapidly decapitated before obtaining spleens, small intestines, or spinal cords. All mouse experiments were performed in accordance with the guidelines of the Institutional Animal Care and Use Committees of the Third Military Medical University.

**Sorting CD4$^+$IL17$^+$ Th17 cells and CD4$^+$IL17$^-$ T cells**. To prepare single-cell suspensions, the spleen and mesenteric lymph nodes of IL17-IRES-EGFP mice (purchased from Beijing Biocytogen Co., Ltd) were mechanically dissociated and passed through 70-μm mesh (BD Biosciences). Splenic CD4$^+$ T cells were purified using an EasySep™ mouse CD4$^+$ T cell isolation kit according to the manufacturer's instructions (STEMCELL Technologies) and then incubated for 30 min at room temperature with an anti-CD4 antibody (1:100, eBioscience, RPA-T4). Subsequently, the CD4$^+$ T cells were sorted into CD4$^+$IL17$^+$ Th17 cells (from an IL17-EGFP mouse) and CD4$^+$IL17$^-$ T cells based on CD4 and GFP expression analyzed by FACS, and the sorting purity was retested with anti-CD4 and anti-RORγt (1:100, eBioscience, AFKJS-9) monoclonal antibodies (mAbs).

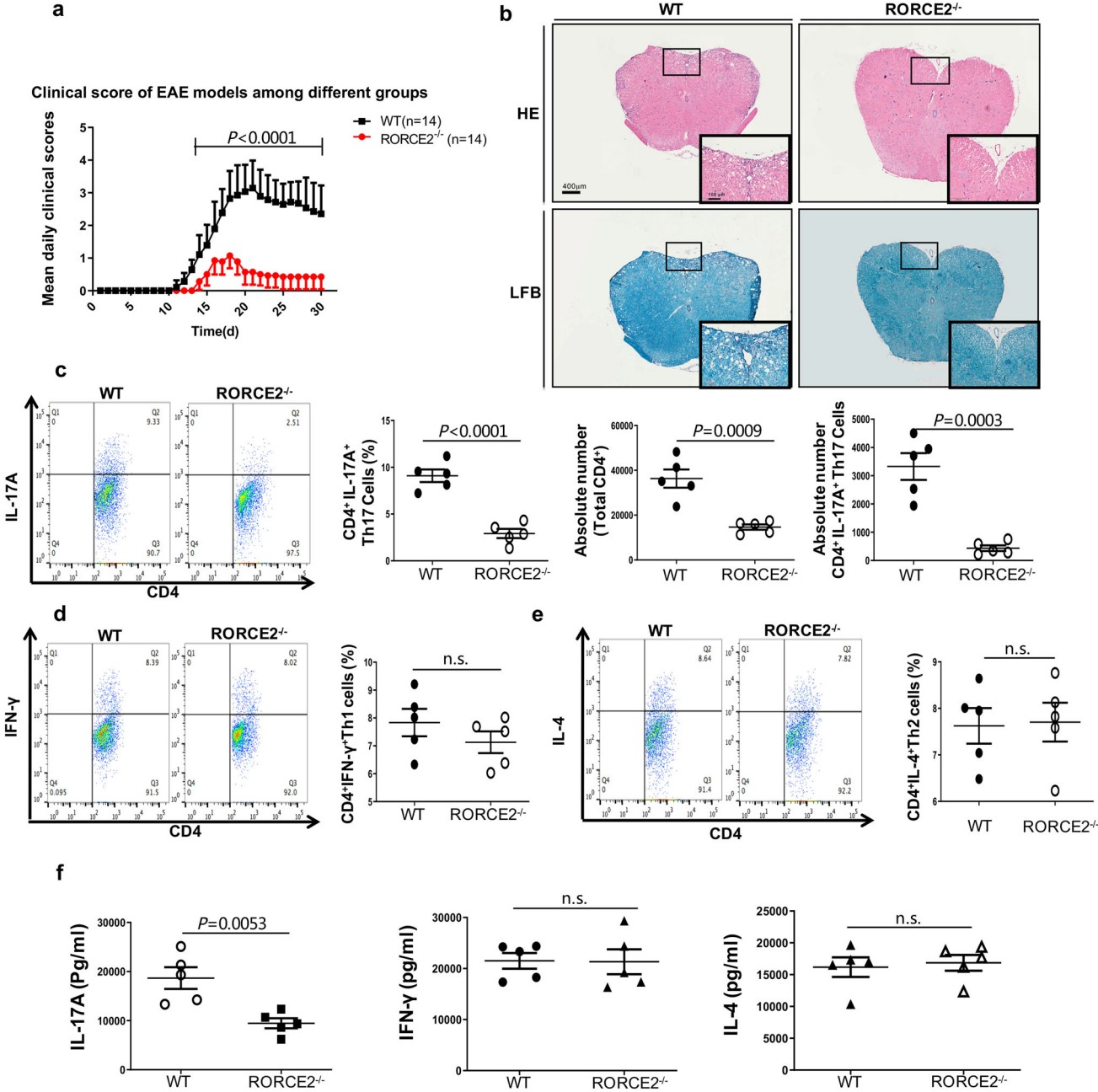

**Fig. 4 RORCE2 deficiency ameliorates neuroinflammation in EAE mice. a** The mean daily clinical scores for RORCE2$^{-/-}$ and WT mice after EAE induction are shown (two-way analysis of variance (ANOVA), $n = 14$ biological independent animals). **b** Spinal cords collected on day 30 from indicated mice were stained with hematoxylin and eosin (HE) or luxol fast blue (LFB) to assess inflammation and myelin content, respectively. The outlines indicate inflammatory or demyelinated foci. Scale bars, 400 or 100 μm (magnified panels). **c** On day 30 after EAE induction, CD4$^+$ T cells among leukocytes isolated from the spinal cord of indicated mice were gated and further analyzed to determine the frequencies of CD4$^+$IL-17A$^+$ Th17 cells. In addition, the absolute numbers of spinal cord-infiltrated CD4$^+$IL-17A$^+$ Th17 cells and total CD4$^+$ T cells were also evaluated by flow cytometry. **d, e** On day 30 after EAE induction, CD4$^+$ T cells among the leukocytes isolated from spinal cord of indicated mice were gated and further analyzed to determine the frequencies of CD4$^+$IFNγ$^+$ Th1 and CD4$^+$IL-4$^+$ Th2 cells by flow cytometry. **f** Mononuclear cells were collected at day 8 from inguinal lymph nodes and further cultured ex vivo with MOG for 3 days. The concentrations of IL-17A, IFNγ, and IL-4 were measured by ELISA, respectively. Mean ± SEM are shown, $n = 5$ biological independent animals, unpaired two-tailed Student's $t$-test (**c–f**). Experimental mice were between 10 and 12 weeks of age, with no preference to gender and were maintained on a C57BL/6 background. Source data are provided as a Source Data file.

**Luciferase reporter assays.** The *RORγt* promoter (−400 to +151 bp relative to the transcriptional start site of *RORγt*, which was defined in a previous study[27]) was amplified by PCR from mouse genomic DNA and then cloned into the pGL3 basic luciferase vector (Promega). Full-length RORCE (RORCEf, chr3: 94,169,746-94,173,149), RORCE1 (chr3: 94,169,746-94,170,945), RORCE2 (chr3: 94,170,946-94,172,113), RORCE3 (chr3: 94,172,215-94,173,149), RORCE1-2 (chr3: 94,169,746-94,172,113), and RORCE2-3 (chr3: 94,170,946-94,173,149) were also amplified by PCR from mouse genomic DNA and then cloned upstream of the *RORγt* promoter or

downstream of the luciferase gene in pGL3 vectors. RORCE2 with STAT3-BS deficiency was synthesized by Qsingke Biotechnology Corp. (Shanghai, China) and cloned upstream of the *RORγt* promoter in the pGL3 vector. All pGL3 clone vectors were verified by sequencing. The PCR primer sequences used for molecular cloning are listed in Supplementary Table 2. To test promoter and enhancer activities, 293T, EL4, and Th17-polarized cells were transfected with the indicated plasmids using Lipo-fectamine 3000 (Invitrogen) or Amaxa Nucleofector reagents (Lonza) according to the manufacturer's instructions and cultured at 37 °C for 24 h in complete RPMI medium.

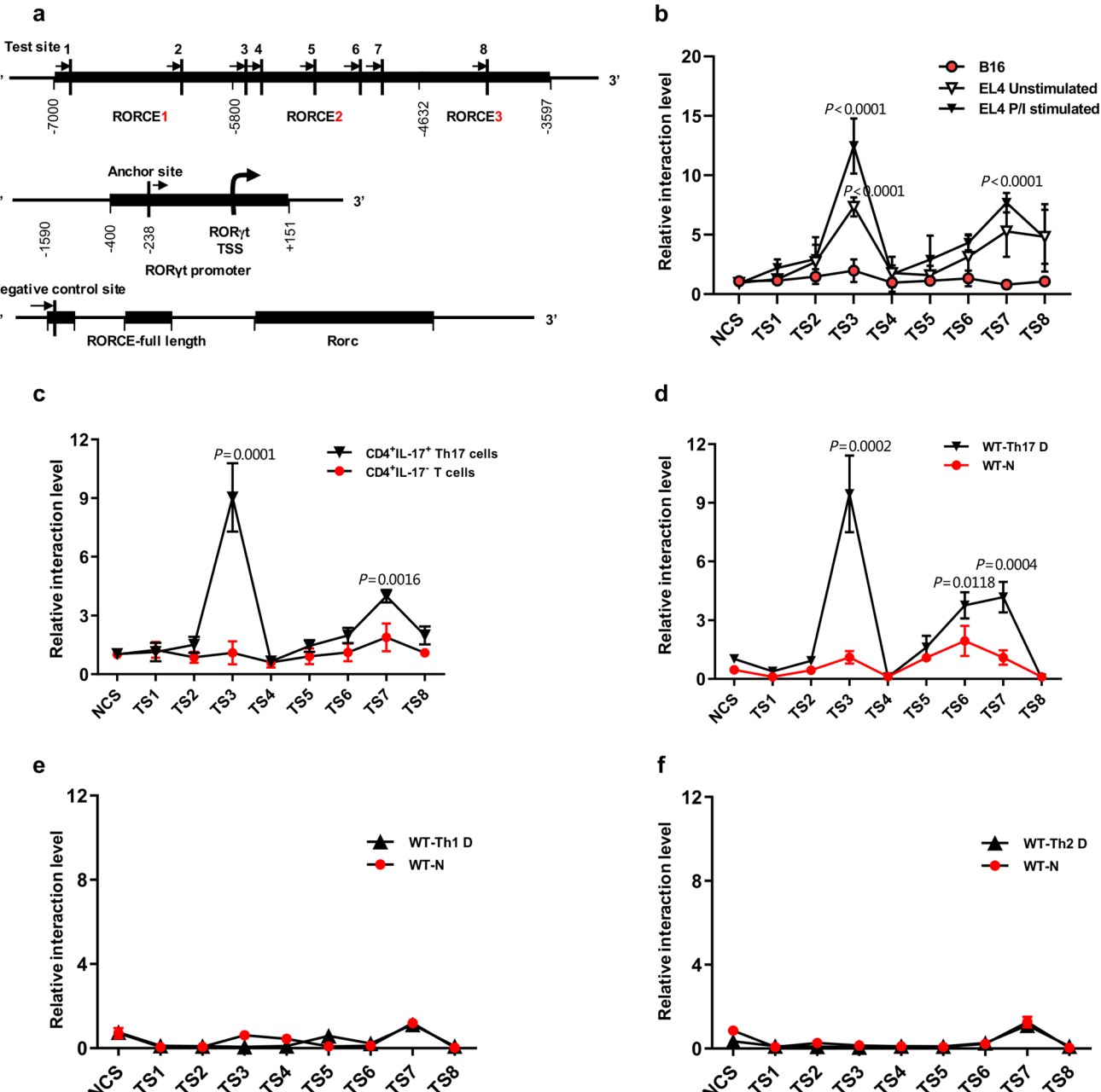

**Fig. 5 3C analysis of the RORCE region. a** Map of the investigated interactions. Since we found NlaIII to be the only restriction enzyme that allowed sufficiently high-resolution analysis of RORCE, NlaIII was finally used to digest the RORCE region. The positions of NlaIII sites (vertical lines) in RORCE used in this 3C analysis are indicated as test sites 1 to 8 (TS1 to TS8). The NlaIII site in $ROR\gamma t$ gene promoter was taken as the anchor site (AS), from which possible interactions with other regions of RORCE were quantified. The primers used for 3C-qPCR are indicated by simple black arrows. The negative control site (NCS) is another NlaIII site upstream of RORCE, which does not interact with the AS. **b–f** Quantitative analysis of a 3C-qPCR assay assessing the RORCE region. The data represent relative interaction frequencies between AS and the other tested sites (TS1 to TS8) in the RORCE region. Relative interaction frequencies were determined by qPCR relative to standard curves as described in the "Methods" section. Data points represent the mean ± SEM for PI-stimulated EL4 (**b**, $n = 4$ independent experiments), FACS-purified CD4+IL17+ Th17 (**c**, $n = 5$ independent experiments), Th17-polarized (**d**, $n = 5$ independent experiments), Th1-polarized (**e**, $n = 5$ independent experiments), and Th2-polarized cells (**f**, $n = 5$ independent experiments) from WT mice. Th17 D, Th1 D, and Th2 D represent Th17, Th1, or Th2 differentiation under the appropriate polarization conditions, respectively, and N represents naïve cells. All P-values are based on unpaired two-tailed Student's t-test (**b–d**). Experimental mice were between 8 and 12 weeks of age, with no preference to gender and were maintained on a C57BL/6 background. Source data are provided as a Source Data file.

After 24 h of transfection, the cells were treated with PMA (50 ng/ml, Sigma-Aldrich) and ionomycin (1 µM, Calbiochem) for 4 h. The cells were harvested, lysed, and assessed with a dual-luciferase reporter system (Promega). The Renilla luciferase activity of pRL-TK (Promega) was used to normalize the transfection efficiency and firefly luciferase activity of the reporter constructs. All values were obtained from experiments performed in triplicate and repeated at least three times. Statistical significance between groups was determined using Student's t-test (GraphPad Prism 8).

**ChIP-qPCR.** Cells were cross-linked using 1% formaldehyde for 10 min at 37 °C and then sonicated (Bioruptor; Diagenode). The DNA–protein complexes were isolated using a ChIP assay kit (Beyotime) according to the manufacturer's instructions with indicated antibodies[44]. The following antibodies were used for ChIP: anti-H3K4me1 (1:250, Abcam, ERP16597), anti-H3K27Ac (1:250, Abcam), anti-SOX-5 (1:250, Abcam, ab94396), and anti-STAT3 (1:250, Cell Signaling Technology, 124H6). Purified DNA was subjected to qPCR analysis using SYBR Green master mix (TakaRa).

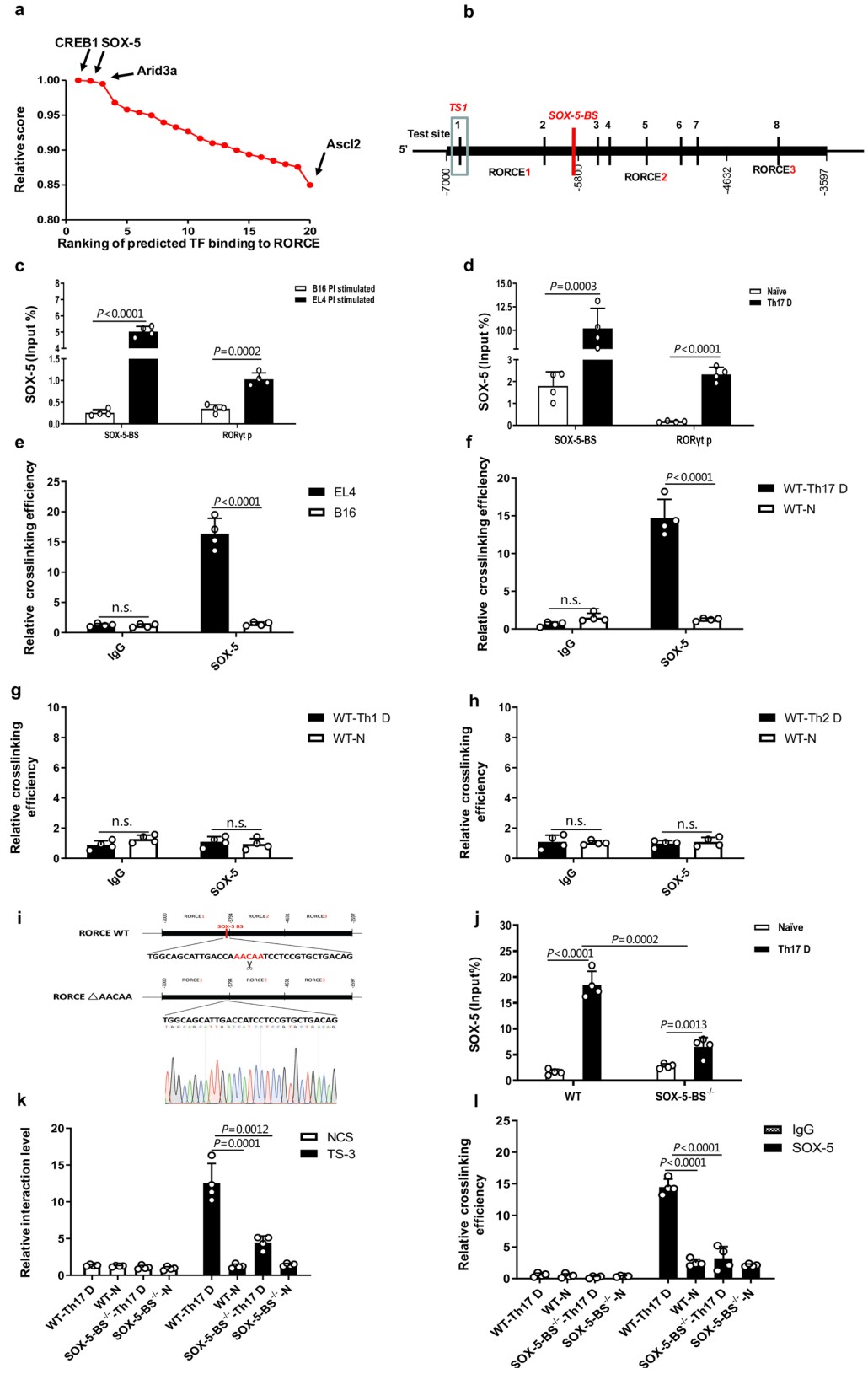

The primers for the ChIP-qPCR analysis are listed in Supplementary Table 3. Data are expressed as the percentage of the input for each ChIP fraction.

**RT-qPCR validation of relative mRNA expression**. Total RNA was isolated from cells or tissues by TRIzol (Invitrogen), and SYBR Green-based qPCR was performed on cDNA first-strand synthesis products generated from the isolated total RNA (TakaRa). The reactions were performed in triplicate. Relative mRNA

expression was quantified using the $\triangle\triangle$Ct method, and glyceraldehyde 3-phosphate dehydrogenase (GAPDH) was used as an internal control. The qPCR primers are listed in Supplementary Table 4.

**Differentiation of CD4[+] T cells in vitro**. Naïve CD4[+] T cells were isolated from 8- to 12-week-old mice with an EasySep™ mouse naïve CD4[+] T cell isolation kit (STEMCELL) according to the manufacturer's instructions. Sorted naïve CD4[+]

**Fig. 6 ChIP-loop analysis of the RORCE region. a** The ranking of predicted transcriptional factors (TFs) capable of binding to RORCE was performed based on relative scores predicted with the JASPAR database (http://jaspar.genereg.net). **b** Red vertical line shows SOX-5 binding site (SOX-5-BS) in RORCE, which is very close to TS3. **c, d** A ChIP-qPCR assay was performed to determine the enrichment of SOX-5 in SOX-5-BS and *RORγt* promoter (RORγt p) in PI-stimulated EL4 cells (**c**) and WT Th17-polarized cells (**d**). **e–h** A ChIP-loop assay with an anti-SOX-5 antibody or control mouse IgG was performed to quantify relative interaction efficiency between the AS and TS3 mediated by SOX-5 in PI-stimulated EL4 (**e**), Th17-polarized (**f**), Th1-polarized (**g**), and Th2-polarized cells (**h**) from WT mice. **i** The sequence of SOX-5-BS (AACAA) in RORCE was deleted by using CRISPR/Cas9 technology (top). The knockout allele was confirmed by sequencing (bottom). **j** A ChIP-qPCR assay was performed to test the enrichment of SOX-5 in SOX-5-BS in Th17-polarized cells from SOX-5-BS-deficient (SOX-5-BS$^{-/-}$) or WT mice. **k, l** Quantitative analysis of 3C-qPCR (**k**) and ChIP-loop assays (**l**) evaluating the interaction between TS3 and AS in Th17-polarized cells from SOX-5-BS-deficient or WT mice was performed. Th17 D, Th1 D, and Th2 D represent Th17, Th1, or Th2 differentiation under the corresponding polarization conditions, respectively, and N represents naïve cells. Mean ± SEM are shown, $n = 4$ independent experiments, unpaired two-tailed Student's $t$-test (**b–h, j–l**). Experimental mice were between 8 and 12 weeks of age, with no preference to gender and were maintained on a C57BL/6 background. Source data are provided as a Source Data file.

T cells were cultured on irradiated splenocytes (2000 rads) with soluble anti-CD3 (2 μg/ml, 145-2C11; BioXCell) at a ratio of 1:5 in a 24-well plate. The naïve cells were cultured at $1.5 × 10^6$/ml in T cell medium, sodium pyruvate, Hepes, penicillin/streptomycin, gentamicin sulfate, and 2-mercaptoethanol. The following cytokines were added to generate Th17 subset: IL-6 (20 ng/ml; Miltenyi Biotec), TGF-β1 (2 ng/ml; Miltenyi Biotec), anti-IL-4 (10 μg/ml, 11B11, BioXCell), and anti-IFNγ (10 μg/ml, XMG1.2, BioXCell). On day 3 after stimulation, we transferred four 24-wells into one 10-cm dish containing 10 ml of T cell medium, IL-6, TGF-β1, anti-IL-4, and anti-IFNγ. This time we supplemented IL-2 (15 U/ml) into the T cell media and cultured the cells for additional 2 days before analysis. Th1 and Th2 polarizations were performed using CellXVivo mouse Th1 cell differentiation kit (R&D Systems, CDK018) and CellXVivo mouse Th2 cell differentiation kit (R&D Systems, CDK019), respectively.

**Flow cytometry and cytokine detection**. Upon harvest, cells were washed and resuspended in phosphate-buffered saline (PBS) containing 2% fetal bovine serum (FBS). For surface staining for CD4, cells were stained with the appropriate antibodies for 30 min at 4 °C. For intracellular staining for TFs including RORγt, TBX21, and GATA3, cells were fixed and permeabilized with Perm/Fix (eBioscience), washed two times with Perm/Wash (eBioscience), and then stained with appropriate antibodies for 30 min in PBS containing 2% FBS. For intracellular staining for cytokines including IL-17A, IL-4, and IFNγ, cells were stimulated with PMA (50 ng/ml, Sigma-Aldrich), ionomycin (1 μM, Calbiochem) for 4 h. We add the protein transport inhibitor Golgi-Stop (BD Bioscience) at a final concentration of 3 μM in the last 2 h of stimulation. The cells were fixed, permeabilized, and then stained with appropriate antibodies for 30 min in PBS containing 2% FBS. Data were acquired on a FACS Canto cell analyzer (BD Bioscience) and analyzed with FlowJo 10.0.7 software (Tree Star). Cell culture supernatants were evaluated by ELISA kits for the secretion of IL-17A, IL-4, and IFNγ (all from Dakewe) according to the manufacturer's instructions.

The following mAbs were used for Th17, Th1, and Th2 cell staining: for surface staining, anti-CD4 (1:100, eBioscience, RPA-T4); for intracellular (cytoplasmic and nuclear) staining, anti-IL-17A (1:100, eBioscience, eBio17B7), anti-IL-4 (1:100, eBioscience, 11B11), anti-IFNγ (1:100, eBioscience, XMG1.2), anti-RORγt, anti-GATA3 (1:100, eBioscience, TWAJ), and anti-TBX21 (1:100, eBioscience, eBio4B10).

**Isolation and analysis of LP lymphocytes**. The intestines from indicated mice were removed, opened longitudinally, and cut into 1 cm pieces. The pieces were then incubated twice in 5 mM EDTA in PBS for 15 min at 37 °C, and then the epithelial cell layer was removed by vortexing and passing through a 100-μm cell strainer. After incubation with EDTA solution, tissues were washed, minced into small pieces, and digested for 1 h at 37 °C in digestion solution containing 4% FBS, 0.5 mg/ml collagenase III (Roche), 0.2 mg/ml DNase I (Sigma-Aldrich), and 2 mg/ml dispase II (Sigma-Aldrich), and lymphocytes were obtained by gradient centrifugation on a 40%/80% Percoll gradient (GE Healthcare)[45]. For flow cytometry analysis, isolated lymphocytes were stained with an anti-lineage cocktail (including anti-CD19 (1:100, eBioscience, 6D5), anti-CD8 (1:100, eBioscience, 53-6.7), anti-Gr1 (1:100, eBioscience, RB6-8C5), and anti-CD3 (1:100, eBioscience, 145-2C11) antibodies) and anti-CD45 (1:100, eBioscience, 30-F11), anti-CD127 (1:100, eBioscience, A7R34), anti-IL-17A, and anti-RORγt antibodies.

**Induction of EAE**. On day 0, 10- to 12-week-old mice were immunized subcutaneously with 200 μg of MOG 33–35 peptide (CHINESE PEPTIDE) emulsified with CFA (Sigma-Aldrich) containing 5 mg/ml heat-killed *Mycobacterium tuberculosis* (BD Bioscience). The mice then received an intraperitoneal injection of 200 ng of pertussis toxin (Millipore) 2 and 26 h after immunization. The mice were monitored daily and scored for clinical signs of disease according to the following criteria: 0 = no clinical symptoms; 1 = limp tail; 2 = weakness in hind limbs; 3 = complete paralysis of hind limbs; 4 = complete hind limb and partial front limb paralysis; and 5 = moribund state[45].

**Isolation and analysis of spinal cord mononuclear cells**. Mice were anesthetized and perfused with cold PBS. The spinal cord was then removed, cut into 0.5-cm pieces, digested with the Neural Tissue Dissociation Kit (MiltenyiBiotec), and homogenized. The mononuclear cells in the spinal cord were isolated by gradient centrifugation at 850×*g* for 30 min on a 40%/80% Percoll gradient. The isolated cells were stimulated with PMA (50 ng/ml, Sigma-Aldrich), ionomycin (1 μM, Calbiochem) for 2 h. Then, we add the protein transport inhibitor Golgi-Stop (BD Bioscience) at a final concentration of 3 μM in the last 2 h of stimulation and then stained with anti-CD45, anti-CD3 (eBioscience), anti-IFNγ (eBioscience), anti-CD4, anti-IL4 (eBioscience), and anti-IL-17A antibodies for flow cytometry analysis.

**3C-qPCR and ChIP-loop assays**. Cells (10M) were washed with PBS twice and were cross-linked with 2% formaldehyde for 10 min at room temperature. The cross-linking was terminated by 0.125 M glycine of for 15 min at 4 °C and then samples were centrifuged at 340×*g* for 8 min. Cross-linked cells were washed with cold PBS and lysed for 30 min with ice-cold lysis buffer containing 10 mM/l Tris (pH 8.0), 10 mM/l NaCl, 0.15% NP-40, and 1 mM/l dithiothreitol (DTT). The nuclei of cells were then harvested and suspended in the appropriate restriction enzyme buffer containing 0.5% SDS and incubated for 1 h at 37 °C with gentle shaking. SDS was then sequestered from the samples by the addition of 1% Triton X-100 for 1 h at 37 °C. The samples were digested with NlaIII restriction enzyme (New England BioLabs (NEB)) for 16 h at 37 °C. Restriction enzymes were inactivated by the addition of 1.6% SDS and further incubation for 25 min at 65 °C. Samples were diluted with T4 DNA ligase buffer (Takara) to achieve ~3 ng of DNA/μl, then 200 U T4 DNA ligase (Takara) was added and incubated for 4 h at 16 °C. Samples were then incubated with 1% Triton X-100 followed by incubation with Proteinase K (200 μg/ml) for 16 h at 65 °C to reverse the cross-linking. This was followed by the addition of 10 μg/ml of RNase and the DNA was purified by phenol–chloroform extraction[29,46–48]. In addition, standard curves for the 3C-qPCR assay were generated from serial dilutions of digested BAC RP23-263K17 (containing the entire genomic region of the *RORγ* gene, which was purchased from Invitrogen). All the 3C products were finally quantified on the CFX Connect™ Real-Time PCR Detection System (Bio-Rad)[47]. To obtain quantification values, the Ct of each 3C sample was first corrected with the parameters of the corresponding standard curve and then normalized to that of a Gapdh loading control ((Value $_{sample}$/ Value $_{Gapdh}$) × 10).

The ChIP-loop assays were performed based on a standard 3C assay[49,50]. The cross-linked chromatin was then immunoprecipitated using an antibody against mouse SOX-5 (1:250, Abcam, ab94396) or control mouse IgG (1:100, Abcam, MOPC-173). The ligation step occurred while the chromatin was still coupled to the beads. The ligation product was analyzed by qPCR as in standard 3C experiments and quantified on the CFX Connect™ Real-Time PCR Detection System (Bio-Rad). The primers for the 3C-qPCR analysis are listed in Supplementary Table 5.

**ChIP-seq analysis**. ChIP-seq libraries of mouse H3K4me2 were mapped onto the mm9 assembly of the mouse genome using Bowtie 2.3.5 (ref. [51]). Only uniquely mapped reads were used for downstream analyses. SICER was used to determine ChIP-enriched regions with the following parameter settings: window size = 200 bp, gap size = 400 bp, and false discovery rate (FDR) = 0.01 (ref. [52]). Browser views of the ChiP-seq data and peaks were generated using Integrated Genomics Viewer (IGV 2.8.13; broadinstitute.org/igv)[53].

**Retrovirus-mediated gene overexpression of Sox5 or Stat3**. Retrovirus-mediated gene overexpression of *Sox5* or *Stat3* in Th17-polarized cells was performed by a RetroNectin-bound virus infection method[30,54]. In brief, we constructed the retrovirus vectors which expressed *Sox5* and *Stat3* genes. Then 48-well plates were coated with RetroNectin (25 μg/ml) and anti-CD3 mAb (BioXCell) overnight at 4 °C. Medium containing retrovirus was added to the RetroNectin-coated plate and the plate was centrifuged for 2 h at 2000×*g* at 32 °C. After washing with PBS, Th17-polarized cells were added to the retrovirus-bound RetroNectin/

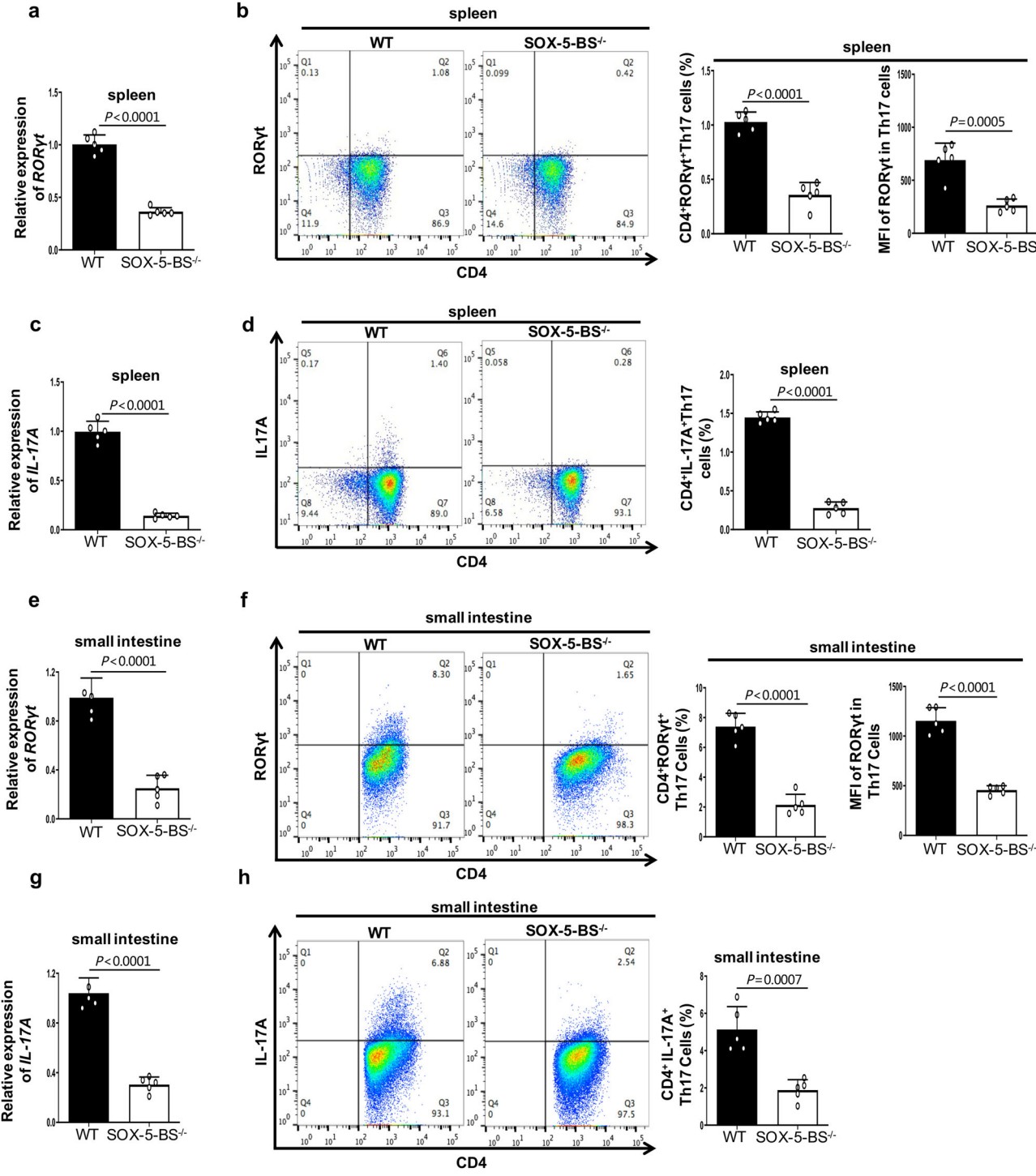

**Fig. 7 SOX-5-BS deletion in RORCE results in reductions in RORγt expression and Th17 cell numbers in vivo. a** Relative mRNA expression of *RORγt* gene in splenic CD4+ T cells from indicated mice. **b** Frequencies of CD4+RORγt+ Th17 cells in splenic CD4+ T cells from indicated mice. The MFI of RORγt in Th17 cells was also analyzed and summarized. **c** Relative mRNA expression of *IL-17A* gene in splenic CD4+ T cells from indicated mice. **d** Frequencies of CD4+IL-17A+ Th17 cells in splenic CD4+ T cells from indicatded mice. **e** Relative mRNA expressions of *RORγt* gene in LP CD4+ T cells from the small intestine of indicated mice. **f** RORγt+ Th17 frequencies in CD4+CD45+ Lin+ lymphocyte population and MFI of RORγt in Th17 cells from LP of small intestine of indicated mice. **g** Relative mRNA expressions of *IL-17A* gene in LP CD4+ T cells from the small intestine of indicated mice. **h** IL-17A+ Th17 frequencies in CD4+CD45+ Lin+ lymphocyte population from LP of small intestine of indicated mice. Mean ± SEM are shown, *n* = 5 independent experiments, unpaired two-tailed Student's *t*-test (**a–h**). Experimental mice were between 8 and 12 weeks of age, with no preference to gender and were maintained on a C57BL/6 background. Source data are provided as a Source Data file.

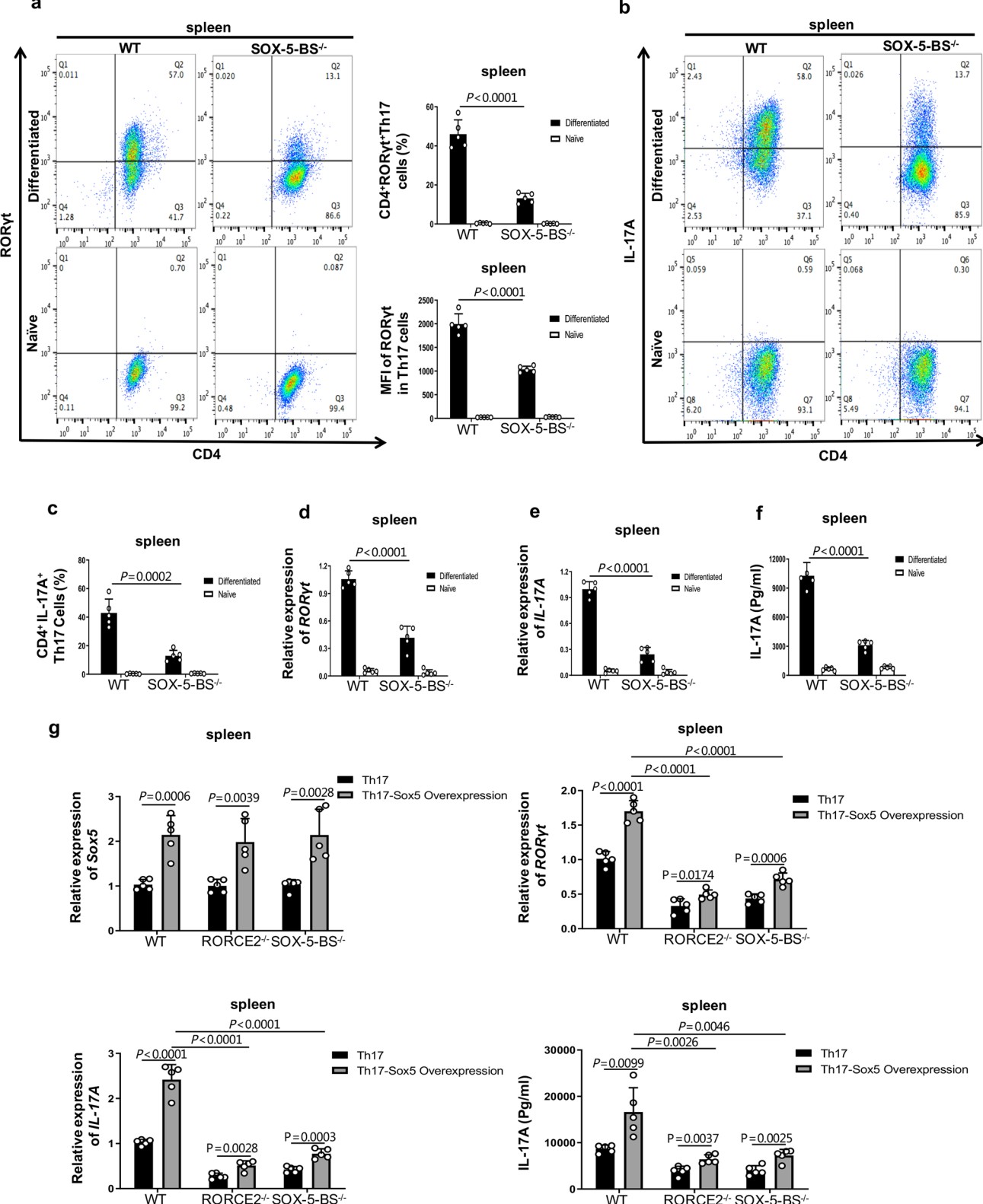

**Fig. 8 SOX-5-BS deficiency in RORCE inhibits Th17 cell polarization in vitro. a–c** Splenic naïve CD4+ T cells from indicated mice stimulated under Th17 polarizing conditions for 4 days. The frequency of CD4+RORγt+ Th17 (**a**) and CD4+IL-17A+ Th17 (**b** and **c**) cells and MFI of RORγt (**a**) were measured by flow cytometry and summarized. **d–f** Relative mRNA expressions of *RORγt* (**d**) and *IL-17A* (**e**) genes in Th17-polarized cells and IL-17A protein in the culture supernatants (**f**) were measured by RT-qPCR and ELISA, respectively. **g** Relative mRNA expression of *Sox5*, *RORγt*, and *IL-17A* genes in Th17-polarized cells from indicated mice and the IL-17A protein in culture supernatants before and after *Sox5* overexpression. Mean ± SEM are shown, *n* = 5 independent experiments, unpaired two-tailed Student's *t*-test (**a**, **c**–**g**). Experimental mice were between 8 and 12 weeks of age, with no preference to gender and were maintained on a C57BL/6 background. Source data are provided as a Source Data file.

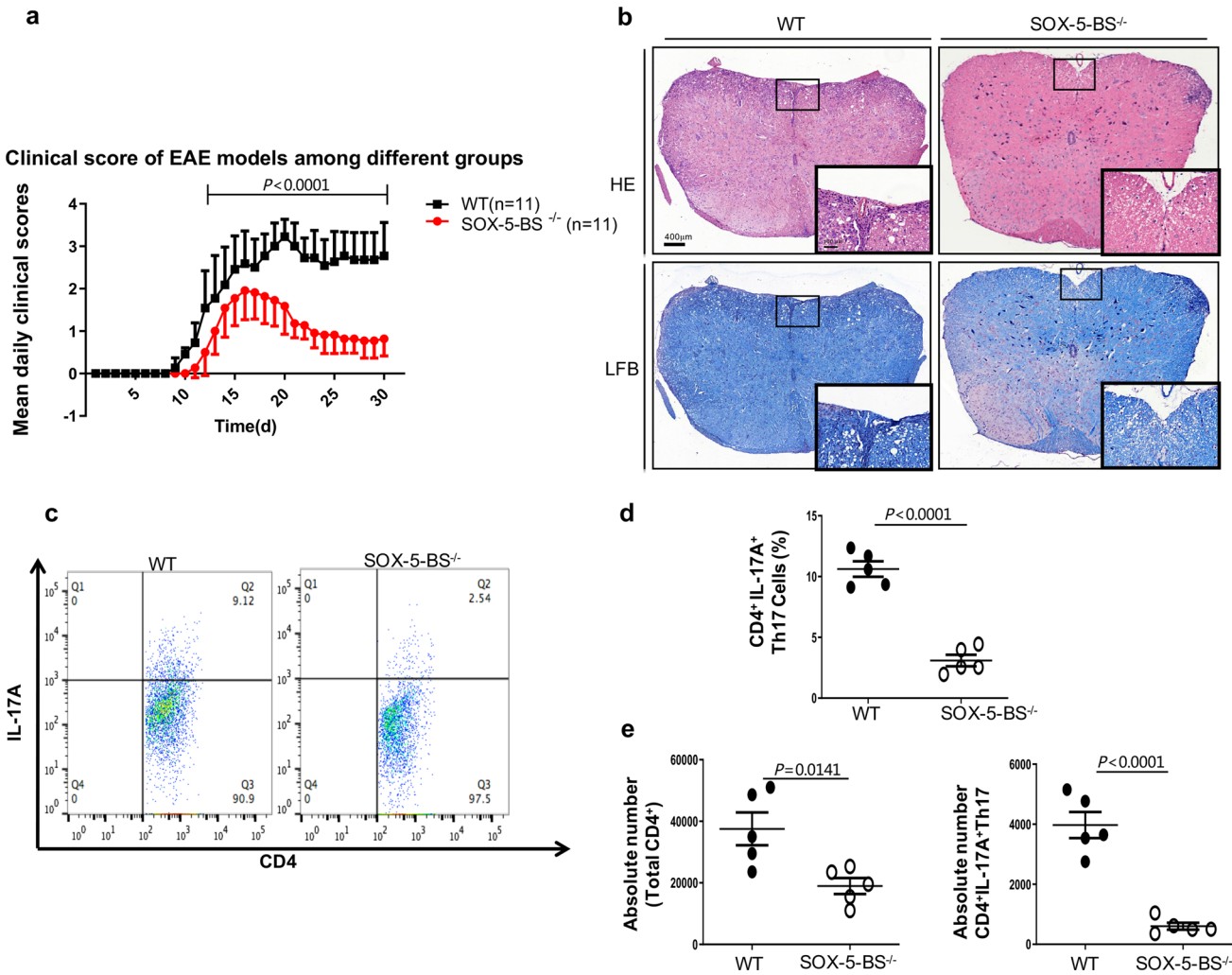

**Fig. 9 SOX-5-BS deletion reduces the severity of EAE. a** Mean daily clinical scores for SOX-5-BS$^{-/-}$ and WT mice after EAE induction are shown (two-way analysis of variance (ANOVA), $n = 11$ biological independent animals). **b** Spinal cords collected on day 30 from indicated mice were stained with hematoxylin and eosin (HE) or luxol fast blue (LFB) to assess inflammation and myelin content, respectively. The outlines indicate the inflammatory or demyelinated foci. Scale bars, 400 or 100 μm (magnified panels). **c, d** On day 30 after EAE induction, CD4$^+$ T cells among the leukocytes isolated from spinal cord of indicated mice were gated and further analyzed to determine the frequencies of CD4$^+$IL-17A$^+$ Th17 cells. **e** Absolute numbers of spinal cord-infiltrated CD4$^+$IL-17A$^+$ Th17 and total CD4$^+$ T cells were evaluated by flow cytometry. Mean ± SEM are shown, $n = 5$ biological independent animals, unpaired two-tailed Student's $t$-test (**d** and **e**). Experimental mice were between 10 and 12 weeks of age, with no preference to gender and were maintained on a C57BL/6 background. Source data are provided as a Source Data file.

anti-CD3 mAb-coated plates in the presence of anti-CD28 mAb (2 μg/ml, BioX-Cell, D665) and were cultured for 48 h at 37 °C. The cells were then harvested for the analysis of target gene expression.

**Stat3 knockdown analysis**. The siRNA for *Stat3* (stealth select RNAiSTAT3-MSS209601) and stealth RNAi negative control (12935100) duplexes were purchased from Invitrogen; 100 pM siSTAT3 or negative controls were transfected to Th17-polarized cells ($5 \times 10^5$ cells) by using Amaxa Nucleofector reagents (Lonza) according to the manufacturer's instructions and cells were cultured in plate bound anti-CD3 mAb in the presence of anti-CD28 mAb for 24 h. The cells were then harvested for the analysis of target gene expression.

**Immunoblotting and Co-IP analysis**. Cell were harvested and lysed on ice in lysis buffer containing 0.5% Triton X-100, 20 mM Hepes pH 7.4, 150 mM NaCl, 12.5 mM β-glycerophosphate, 1.5 mM MgCl₂, 10 mM NaF, 2 mM DTT, 1 mM sodium orthovanadate, 2 mM EGTA, 20 mM aprotinin, and 1 mM phenylmethylsulfonyl fluoride for 30 min, followed by centrifuging at 12,000 r.p.m. for 15 min to extract clear lysates. For Co-IP, cell lysates were incubated with 4 μg of antibody at 4 °C overnight, followed by incubation with G-sepharose beads for 2 h, and the beads were washed five times with lysis buffer and the precipitates were eluted with 2× sample buffer[55]. Elutes and whole-cell extracts were resolved on SDS-PAGE followed by immunoblotting (IB) with indicated antibodies. The following vectors

were used: pcDNA3.1-SOX-5 Flag which expressed Flag-tagged SOX-5 and pcDNA3.1-STAT3 HA which expressed HA-tagged STAT3. In brief, Hela cells were transfected with pcDNA3.1-SOX-5 Flag, together with pcDNA3.1-STAT3 HA by Lipofectamine 3000 Transfection Kit (Invitrogen) and then cultured in plate for 48 h. After stimulated with PI and Biochanin A (10 μM)[56], whole-cell lysates from Hela cells were subjected to IP with anti-HA antibody (1:250, Abcam, HA.C5) and IB with anti-HA (1:1000, Abcam, HA.C5) or anti-Flag (1:1000, Sigma-Aldrich, M2). Input proteins (input) were also immunoblotted with anti-HA or anti-Flag antibody. Furthermore, Th17 cells that were differentiated from naïve CD4$^+$ T cells under Th17-polarization condition were also subjected to IP with anti-STAT3 antibody (1:100, Cell Signaling Technology, 124H6) or control mouse IgG (1:100, Abcam, MOPC-173) and IB with anti-STAT3 or anti-SOX-5 antibody. Images of IB experiments were collected via FUSION solo-s (VILBER).

**Statistical analysis**. Statistical analyses were performed using GraphPad Prism version 8.0 software. Results are shown as mean and the error bar which represents SEM of biological replicates as indicated in the figure legends. Non-parametric statistics was applied to compare differences between two groups. Statistical significance was determined by unpaired Student's $t$-test (two-tailed) for two groups. For the EAE model, changes in clinical scores were compared using two-way analysis of variance (ANOVA). Significance of data is denoted by the exact $P$-values (unless $P < 0.0001$) which are shown in the figures. $P > 0.05$ is considered non-significant and denoted as n.s.

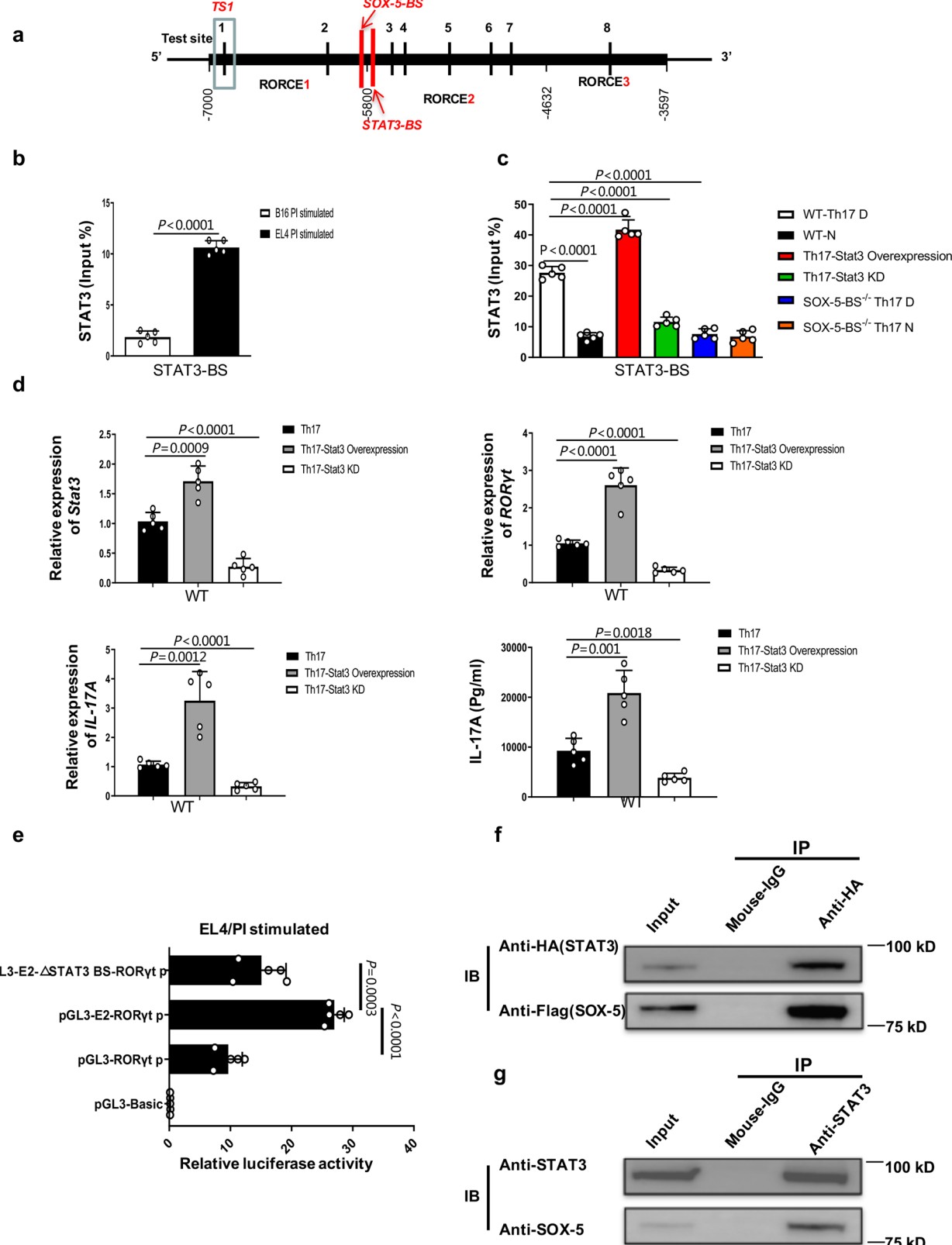

**Reporting summary**. Further information on research design is available in the Nature Research Reporting Summary linked to this article.

## Data availability

We downloaded the mouse H3K4me2 ChIP-seq data, including GSM1032374, GSM1032375, GSM2258669, GSM2258675, and GSM2258681, from the GEO database. We also used the Roadmap Epigenomics Project database (http://egg2.wustl.edu/roadmap/web_portal/) to analyze the chromatin signature of a region similar to mouse RORCE (indicated by the red box) including H3K27me3, H3K9me3, H3K27Ac, H3K36me3, H3K4me1, and H3K4me3 in human Th17 and Th0 cells. We also used the JASPAR database to predict the transcriptional factor binding sites in RORCE (http://jaspar.genereg.net/). All the other data supporting the findings of this study are available within the article and its supplementary information files or available from

**Fig. 10 STAT3 is necessary for the activation of RORCE2. a** Red vertical line shows STAT3 binding site (STAT3-BS) in RORCE2, which is located approximately 50 bp downstream of the SOX-5-BS. **b, c** A ChIP-qPCR assay was performed to determine the enrichment of STAT3 in its binding site in RORCE2 in EL4 cells (**b**), WT Th17-polarized cells before and after STAT3 overexpression or knockdown (KD) and Th17-polarized cells generated from SOX-5-BS-deficient mice (**c**). Th17 D represents Th17-polarized cells, and N represents naïve cells. **d** Relative expression of *Stat3*, *RORγt*, and *IL-17A* genes in WT Th17-polarized cells and IL-17A production in culture supernatants before and after *Stat3* overexpression or knockdown (Stat3 KD) were determined by RT-qPCR and ELISA, respectively. **e** Luciferase assays were performed with different reporter constructs in EL4 cells. RORCE2 with a STAT3-BS deletion (E2-ΔStat3BS) was cloned upstream of *RORγt* promoter in the pGL3 vector, and transcriptional activity was assessed with a dual-luciferase system. **f** Hela cells were transfected with pcDNA3.1-SOX-5 Flag vector together with pcDNA3.1-STAT3 HA vector and then cultured for 48 h. After stimulated with PI and Biochanin A, whole-cell lysates from Hela cells were subjected to IP with anti-HA antibody or control mouse IgG, and IB with anti-HA or anti-Flag HRP antibody. Input proteins (input) were also IB with anti-HA or anti-Flag HRP antibody. **g** Whole-cell lysates from Th17 cells were subjected to IP with anti-STAT3 antibody or control mouse IgG and IB with anti-SOX-5 or anti-STAT3 antibody. Input proteins (input) were also IB with anti-SOX-5 or anti-STAT3 antibody. Mean ± SEM are shown, $n = 5$ independent experiments, unpaired two-tailed Student's *t*-test (**b–e**). Experimental mice were between 8 and 12 weeks of age, with no preference to gender and were maintained on a C57BL/6 background. Source data are provided as a Source Data file.

the corresponding author upon reasonable request. Source data are provided with this paper.

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

## Acknowledgements

We thank Dr. Jun Zhu (Mokobio Biotechnology R&D Center) for reading and revising the paper. This work was supported by grants from the National Key Research and Development Project (Nos. 2016YFA0502203 and 2016YFA0502204) and the National Natural Science Foundation of China (Nos. 31670889 and 31200668). The funder had no role in the study design, data analysis, or decision to publish.

## Author contributions

Y.T., B.N., and Y.W. conceived and supervised this study. C.H., Z.W., X.S., J.Z., and H.D. performed the experiments. X.F., Z.T., D.Y., Y.S., J.Y., and S.W. analyzed the data. Y.C. and C.H. contributed to the experimental design related to the knockout mouse model. Y.T., Y.W., and B.N. wrote the manuscript.

## Competing interests

The authors declare no competing interests.
