## [Peer Review File · Nature Communications]

Reviewers' comments:

Reviewer #1 (Th17, transcriptional regulation)(Remarks to the Author):

In their manuscript, "Sox5 activates the RORgt enhancer to facilitate experimental autoimmune encephalomyelitis by promoting Th17 cell differentiation", Tian et al. reported the discovery of a novel RORCE2 enhancer, located -5.8 kb to -4.6 kb upstream of the Rorc promoter. RORCE2-deficient mice (generated by Crispr/Cas9) lacked endogenous RORgt+ Th17 cells [although RORgt+ ILC3 were spared] and were resistant to experimental autoimmune encephalomyelitis (EAE) model. By using a chromosome conformation capture-qPCR (3C-qPCR), the authors established that the RORCE2 enhancer was involved in the formation of a chromatin loop with the Rorc promoter. Furthermore, the authors demonstrated that the transcription factor Sox5 was involved in the process of chromatin loop formation between the RORCE2 enhancer and the Rorc promoter because genetic deletion of the Sox5-binding site in the RORCE2 enhancer or Sox5 gene deficiency abrogated the loop formation, RORgt expression and Th17 cell differentiation. Similarly to RORCE2-deficient mice, Sox5-deficient mice were also resistant to EAE. Since Sox5 lacks DNA transactivation domain, the authors searched for the Sox5-interacting protein at the RORCE2 enhancer. They identified a STAT3-binding site in the vicinity of the Sox5-binding site and showed that the deletion of Sox5-binding site in the RORCE2 element significantly diminished STAT3 binding to the RORCE2 enhancer and abolished Rorc expression.

The authors have carried out a methodical and detailed molecular analysis of the Rorc locus and provided experimental evidence for their conclusions.

Minor questions:

1. Figure 1C – RORCE2 element when cloned into the Rorc promoter-driven luciferase construct demonstrated enhancer activity. However, RORCE2 cloned with either neighboring RORCE1 or RORCE3 element had no enhancer activity. Since all three RORCE elements are present in the endogenous Rorc locus, how the authors explain this finding?
2. Do STAT3 and Sox5 interact? Although Sox5-binding site in the RORCE2 element is required for STAT3 binding, is Sox5 protein required for STAT3 binding to the RORCE2 element?

Reviewer #2 (Th17, transcriptional regulation)(Remarks to the Author):

In the manuscript entitled "Sox5 activates the RORgt enhancer to facilitate experimental autoimmune", the authors identified a putative enhancer in the Rorc gene locus – RORCE2, which can regulate RORgt expression in Th17 cells both in vivo and in vitro. By using 3C-qPCR and ChIP-loop assay, the authors showed that RORCE2 looped with the Rorc promoter, likely dependent on SOX5. Mutating the putative SOX5 binding site abolished STAT3 binding to RORCE2, reduced RORgt expression and resulted in alleviated diseases in EAE models, which mimics the deficiency of RORCE2. The findings in this study, though might be novel and interesting, the mild effect of RORCE2-deficiency on RORgt expression in vitro (Fig 3A) and in vivo (Fig 4C) system clearly could not justify its drastic effect on EAE phenotypes. In addition, IL-17 staining data were missing in many of their experiments, making it hard to judge the importance of RORCE2 in Th17 cell differentiation. Moreover, the SOX5-related mechanism needs to be reexamined and reconsidered, as the reviewer simply could not identify a role (even a very minor effect) of Sox5 in Th17 cell differentiation in the lab, though a previous publication suggested such a role.

Major concerns:

1. Proper Figure #s should be put on each of the Figures.
2. Figure 3, the authors only detected IL-17A expression in the culture supernatant, it should be examined by flow staining for protein expression and real time PCR for mRNA expression.
3. Figure 4, considering the minor defect of ROR γ t expression following RORCE2 expression in in vitro and in vivo (Fig 3A/4C), the great EAE phenotype in Figure 4A is unlikely caused by RORCE2-deficiency, the authors should consider alternative possibility, since CRISPR-Cas9 technology is prone to cause off target deletions or mutations.
4. Figure 4, the authors showed staining data of Th1 and Th2 cells in the EAE model, but where was the IL-17A staining data?
5. Figure 4, the authors had 9 mice in each group in the EAE experiments, but showed less and variable number (3-5) of mouse data for statistic analysis in 4C/D. A reasonable explanation should be given.
6. Figure 7/8, again, the authors missed the IL-17A staining data in SOX5 BS-KO T cells and in the EAE models. It has to be mentioned that the reviewer had evaluated the role of Sox5 in Th17 cells in numerous in vitro culture system, and did not found any effect of Sox5-deficiency on IL-17 expression in Th17 cells in past studies.
7. Figure 9C, according to previous published STAT3 ChIPseq data (Ciofani M et al., Cell, 2012), RORCE2 is not a major STAT3 binding site. It is unclear about the functional relevance of STAT3-RORCE2 interaction, and it is also unclear how SOX5 affected STAT3 recruitment.
8. The authors suggested that RORCE2 regulated ROR γ t expression and Th17 cell differentiation through interacting with SOX5, more direct evidences should be provided. For example, testing the effect of Sox5 overexpression on ROR γ t and IL-17 expression in WT, RORCE2-deficient and SOX5-BS mutant Th17 cells.

Minor concerns:

1. It should be cautious to call RORCE2 as an enhancer, as the authors did not have any evidences that RORCE2 is able to drive Rorc or irrelevant gene expression or promoter activities in a position-independent manner.
2. Figure 2B-G, Figure 7A-D, splenic CD4+ T cells are not ideal for detecting ROR γ t expression, due to the very low ROR γ t expression. Gut tissue would be much better.
3. Figure 1C-D/9D, luciferase activity of RORCE2-construct should be shown in primary Th17 cells, if possible, as well as in Th1 and Th2 cells
4. Figure 4C, the total average CD4+ T cells were ~30000 in WT mice, the ROR γ t+Th17 cells occupied ~10% total CD4+ T cells, it is unclear how to make an absolute number of ~20000 for CD4+ROR γ t+Th17 cells in their assay.

Reviewer #3 (Chromatin structure/architecture)(Remarks to the Author):

The identification of TH17-specific enhancers is a rather interesting topic due to intensive investigation of TH17-dependent pathologies. The authors, based on publically available data tried to identify new enhancer elements for the regulation of the ROR(γ)t transcription factor. They presented a plethora of research approaches including both biochemical in vitro as well as in vivo approaches. It is noteworthy that in support of their working hypotheses, the authors created and utilised many genetically modified mice or cell lines.

In certain parts of the manuscript there are overstatements that stem from the wrong interpretation of the data (such as PCR of genomic fragments that are deleted).

Moreover certain immunological assays seem not to be properly working (such as TH17 cell differentiation).

I have the following comments for the authors:

- Figure 1B: it is not defined what the authors mean by ROR γ t promoter. By what means is the gene's promoter defined?
- Figure 1C/1D: It is an overstatement to talk about the significant effect of a potential enhancer when the proximal promoter transcriptional activity is increased by 2-3 fold maximum.
- The statement of results part #1 regarding the identification of a TH17-specific enhancer is not supported by the data. Only transient transfection assays have been performed in EL4 and 293T cells (the cell types are not discussed at all, by the way) which cannot support the TH17-specific function of a potential regulatory element (that increases basic promoter activity by 3-fold) and assess its importance in TH17 cell differentiation and function.
- In Figure 2C it is not clear how TH17 cells have been isolated for the intracellular staining of ROR γ t. For Fig2C/2D and others the authors do not refer to any quantitative differences in the results section but only refer to qualitative differences which they overestimate.
- I assume that the authors in line 135 mean IL-4 in TH2 cells and IFN γ in TH1 cells.
- There is no information provided regarding the nature of utilized cell lines when first cited. For example, what is the origin of the B16 cell line which is presented completely different from the EL4 cell line.
- In Figure 5, why do the authors used primers of opposite orientation of the anchor site and the test sites. Traditionally same orientation primers are utilized.
- Lines 169-170. The statement regarding the loss of loopscape structure due to the RORCE2 deletion is wrong. The loop is not lost, it is simply the DNA sequence deleted and therefore the PCR primers cannot anneal and amplify the respective regions. (The 3C PCR primers designed for the amplification of ROCE2 anneal in a DNA region from -5800 up to -4632. The knockout mouse bears a deletion from -5793 up to -4630).
- The same comment applies for the statement in lines 170-173 regarding the loss of loop formation in TH1 and TH2-polarised cells stemming from the KO mouse.
- The same comment applies for the statement in lines 186, 188-189 regarding the loss of loop formation in the ChIP-loop experiment with a SOX5 antibody in the RORCE2 deletion.
- In supplemental Figure S5, it is not clear to me therefore I would like a comment from the authors regarding the high levels of Sox5 expression in EL4 cell without any stimulation (since these cells are not TH17 cells) while in T cells differentiated under neutral conditions Sox5 is not expressed (Fig.S5 left panel).
- Figure 6C: please correct ROR γ t p (γ).
- In Figure 7 there seems to be an inherent problem of the whole FACS analysis. It is not clear whether the TH17 polarization conditions are good enough to provide cells of the TH17 cell lineage. The percentages of TH17 cell even in a wt genetic background are too low.

- It is not clearly articulated what is the relationship of SOX5 and STAT3 proteins. Do they physically interact? Do they bind to nearby motifs synergistically?

Reviewers' comments:**Reviewer #1 (Th17, transcriptional regulation)(Remarks to the Author):**

In their manuscript, “Sox5 activates the ROR γ t enhancer to facilitate experimental autoimmune encephalomyelitis by promoting Th17 cell differentiation”, Tian et al. reported the discovery of a novel RORCE2 enhancer, located -5.8 kb to -4.6 kb upstream of the Rorc promoter. RORCE2-deficient mice (generated by Crispr/Cas9) lacked endogenous ROR γ t⁺ Th17 cells [although ROR γ t⁺ ILC3 were spared] and were resistant to experimental autoimmune encephalomyelitis (EAE) model. By using a chromosome conformation capture-qPCR (3C-qPCR), the authors established that the RORCE2 enhancer was involved in the formation of a chromatin loop with the Rorc promoter. Furthermore, the authors demonstrated that the transcription factor Sox5 was involved in the process of chromatin loop formation between the RORCE2 enhancer and the Rorc promoter because genetic deletion of the Sox5-binding site in the RORCE2 enhancer or Sox5 gene deficiency abrogated the loop formation, ROR γ t expression and Th17 cell differentiation. Similarly to RORCE2-deficient mice, Sox5-deficient mice were also resistant to EAE. Since Sox5 lacks DNA transactivation domain, the authors searched for the Sox5-interacting protein at the RORCE2 enhancer. They identified a STAT3-binding site in the vicinity of the Sox5-binding site and showed that the deletion of Sox5-binding site in the RORCE2 element significantly diminished STAT3 binding to the RORCE2 enhancer and abolished Rorc expression.

The authors have carried out a methodical and detailed molecular analysis of the Rorc locus and provided experimental evidence for their conclusions.

Minor questions:

1. Figure 1C – RORCE2 element when cloned into the Rorc promoter-driven luciferase construct demonstrated enhancer activity. However, RORCE2 cloned with either neighboring RORCE1 or RORCE3 element had no enhancer activity. Since all three RORCE elements are present in the endogenous Rorc locus, how the authors explain this finding?

Response:

We appreciate the reviewer’s helpful comments. The previous results were obtained from the EL4 and 293T cell lines, which might not exactly reflect the activity of the candidate enhancers in Th17 cells because there may be several Th17 cell-specific factors contributing to the activity of the candidate enhancers. So, to address the reviewer’s concerns, we further detected the luciferase activity of these constructs containing RORCE1, RORCE2, RORCE3 and their combinations upstream of ROR γ t promoter in Th17-polarized cells. Results showed that all the constructs containing RORCE2 were capable of enhancing the luciferase activity, compared with the controls (new Figure 1 E). When these fragments were cloned downstream of the luciferase gene in the reporter constructs, these constructs also showed a similar luciferase activity in Th17-polarized cells (new Figure 1 F). These results indicated that RORCE2 might act as a potential enhancer for ROR γ t gene in a position-independent manner.

However, we found that the activities of E1-E2, E2-E3 or full length RORCE seem still lower than that of E2. We cannot explain well these observations, but we can speculate that the

steric hindrance effect of E2 flanking fragments may affect E2 activity. In addition, the luciferase reporter system is helpful for the screening of potential enhancers, which however could not accurately reflect the *in vivo* status of certain enhancers. The key results should be from the *in vivo* E2 KO model. Importantly, our *in vivo* E2 KO mice indeed showed the remarkable effects of E2 on the expression of ROR γ t gene. We hope such explanation could address the reviewer's concerns.

2. Do STAT3 and Sox5 interact? Although Sox5-binding site in the RORCE2 element is required for STAT3 binding, is Sox5 protein required for STAT3 binding to the RORCE2 element?

Response:

We appreciate the reviewer for the comments. In the revised manuscript, we conducted Co-IP assay and found that STAT3 interacted with Sox5 (new Figure 9F). Therefore, Sox5 might recruit the STAT3 to RORCE2 by this interaction and the knockout of the Sox5-binding site (i.e., Sox5 protein loss in RORCE) resulted in the loss of STAT3 binding to RORCE2, suggesting that Sox5 protein is required for STAT3 binding to the RORCE2.

Reviewer #2 (Th17, transcriptional regulation)(Remarks to the Author):

In the manuscript entitled “Sox5 activates the ROR γ t enhancer to facilitate experimental autoimmune”, the authors identified a putative enhancer in the *Rorc* gene locus – RORCE2, which can regulate ROR γ t expression in Th17 cells both *in vivo* and *in vitro*. By using 3C-qPCR and ChIP-loop assay, the authors showed that RORCE2 looped with the *Rorc* promoter, likely dependent on SOX5. Mutating the putative SOX5 binding site abolished STAT3 binding to RORCE2, reduced ROR γ t expression and resulted in alleviated diseases in EAE models, which mimics the deficiency of RORCE2. The findings in this study, though might be novel and interesting, the mild effect of RORCE2-deficiency on ROR γ t expression *in vitro* (Fig 3A) and *in vivo* (Fig 4C) system clearly could not justify its drastic effect on EAE phenotypes. In addition, IL-17A staining data were missing in many of their experiments, making it hard to judge the importance of RORCE2 in Th17 cell differentiation. Moreover, the SOX5-related mechanism needs to be reexamined and reconsidered, as the reviewer simply could not identify a role (even a very minor effect) of Sox5 in Th17 cell differentiation in the lab, though a previous publication suggested such a role.

Major concerns:

1. Proper Figure #s should be put on each of the Figures.

Response: Done.

2. Figure 3, the authors only detected IL-17A expression in the culture supernatant, it should be examined by flow staining for protein expression and real time PCR for mRNA expression.

Response: Done (new Figure 3D, 3E and 3H).

3. Figure 4, considering the minor defect of ROR γ t expression following RORCE2 expression in *in vitro* and *in vivo* (Fig 3A/4C), the great EAE phenotype in Figure 4A is unlikely caused by RORCE2-deficiency, the authors should consider alternative possibility, since CRISPR-Cas9 technology is prone to cause off target deletions or mutations.

Response:

We thank the review for the key comments on the CRISPR-Cas9 off-target issue. To address the reviewer's concerns, we have further detected the top 10 of potential off-targets of upstream/downstream sgRNAs in RORCE2 KO mice and found that there were not deletions or mutations. These results are shown in the **new Figure S3**.

As for Figure 3A, keeping in mind the reviewer's concern about the inconsistency between the results of Fig 3A/4C and Fig 4A, we further optimized the protocol of *in vitro* differentiation of Th17 cells. Specifically, we further optimized the cell density of initial naïve CD4⁺ T cells, from 2×10⁶/ml to 1×10⁶/ml, according to the protocol of Th17 differentiation kit and the published literatures [J Exp Med. 2014 Aug 25;211(9):1857-74] because we found that 1×10⁶/ml of initial naïve CD4⁺T cells led to much less non-Th17 proliferation, compared with 2×10⁶/ml of initial naïve CD4⁺T cells. We further optimized the time of Th17 polarization from 5 days to 4 days, because the 5 days caused more cell death than 4 days in our experimental system; more importantly, 4 days are enough for the Th17 polarization as supported by the published literatures [J Immunol. 2014 Jan 1;192(1):110-22; Nat Commun. 2019 Nov 26;10(1):5450]. We ultimately found that there was more than 70% decrease for the frequency of Th17 in KO mice, in contrast to the previous results (less than 30% decrease) in KO mice. These results are shown in the **new Figure 3A-3H**. We thank the reviewer's helpful comments once more and the revised results according to the reviewer's suggestions have further strengthened the conclusion of this study.

For Figure 4C, according to the reviewer's suggestion, we further detected the frequency of CD4⁺IL-17A⁺Th17 (as shown in **new Figure 4C**). We interestingly found that the difference between the WT and KO mice was more significant than before. This may reflect the difference of staining efficiency between different antibodies (anti-IL-17A vs anti- ROR γ t). We thus replaced the previous CD4⁺ ROR γ t⁺Th17 results with CD4⁺IL-17A⁺Th17 results, the latter would reflect the Th17 function better because IL-17A is an effector of Th17 cells.

In addition, to confirm the EAE phenotype in Figure 4A, we further observed clinical score in additional five EAE mice (now total 14 mice), and found the similar results that have now been added to the **revised Figure 4A**.

4. Figure 4, the authors showed staining data of Th1 and Th2 cells in the EAE model, but where was the IL-17A staining data?

Response:

According to the suggestion, we added the IL-17A staining data in the **revised Figure 4C** as indicated in the above comment.

5. Figure 4, the authors had 9 mice in each group in the EAE experiments, but showed less and variable number (3-5) of mouse data for statistic analysis in 4C/D. A reasonable explanation should be given.

Response:

We apologize for our carelessness for Figure 4C/D results. Both of the numbers of mice in Figure 4C and 4D are actually 5.

Due to the difficult isolation of spinal cord mononuclear cells, we only got the FCM data of 5 mice from 9 mice, which resulted in the different number in Figure 4C/D (5 mice) together with that in Figure 4A (9 mice). We have now corrected this issue in the revised manuscript, and further modified the results in Figure 4C/D and Figure 4A by adding the data from additional 5 mice as described in the Response to reviewer's Comment 3 above.

6. Figure 7/8, again, the authors missed the IL-17A staining data in SOX5 BS-KO T cells and in the EAE models. It has to be mentioned that the reviewer had evaluated the role of Sox5 in Th17 cells in numerous in vitro culture system, and did not found any effect of Sox5-deficiency on IL-17A expression in Th17 cells in past studies.

Response:

According to the reviewer's suggestion, we have conducted the IL-17A staining in Sox5 BS-KO T cells and in the EAE model, and the results have been added in the revised Figure 7 I-K and Figure 8C-D. In addition, to confirm the EAE phenotype in Figure 8A, we further observed clinical score in additional 5 EAE mice (now total 11 mice), and found the similar results that have now been added to the revised Figure 8A.

As for the role of Sox5 in Th17, we had actually repeated these experiments for several times and confirmed that our results were highly reproducible. In addition, in this study, we specifically checked the role of Sox5 in RORCE region rather than the total role of Sox5 in Th17 cells. During the design of this study, we had considered that Sox5 should have other target genes in Th17 cell, thus we determined to delete the Sox5 BS by CRISPR-Cas9 method. Our study strategy is different from that of the reviewer's studies, which thus probably causes different results.

7. Figure 9C, according to previous published STAT3 ChIPseq data (Ciofani M et al., Cell, 2012), RORCE2 is not a major STAT3 binding site. It is unclear about the functional relevance of STAT3-RORCE2 interaction, and it is also unclear how SOX5 affected STAT3 recruitment.

Response:

We appreciate the reviewer's helpful comments on the issue of Sox5-STAT3-RORCE2 interaction. To address the raised concerns, we overexpressed and knocked down STAT3 in WT Th17-polarized cells. We found that STAT3 overexpression led to the increased STAT3-RORCE2 interaction as evidence by ChIP-qPCR (modified Figure 9C), and the up-regulation of ROR γ t and IL-17A expressions as evidence by RT-qPCR and/or ELISA in WT Th17-polarized cells (new Figure 9D). Accordingly, STAT3 knockdown caused the reverse results (new Figure 9D). In addition, loss of STAT3-RORCE2 interaction in RORCE2-deficient Th17 cells resulted in markedly decreased expression of ROR γ t and IL-17A (modified Figure 2 and 3). These results suggest that STAT3-RORCE2 interaction might be important for Th17 cells.

Regarding to the relationship between Sox5 and STAT3, we further conducted the Co-IP assay and found the interaction between the STAT3 and Sox5, suggesting that the Sox5 might affect STAT3 recruitment (new Figure 9F).

8. The authors suggested that RORCE2 regulated ROR γ t expression and Th17 cell differentiation through interacting with SOX5, more direct evidences should be provided. For example, testing the

effect of Sox5 overexpression on ROR γ t and IL-17A expression in WT, RORCE2-deficient and SOX5-BS mutant Th17 cells.

Response:

We agree with the reviewer's comments. We verified that Sox5 overexpression caused significant up-regulation of the ROR γ t and IL-17A expressions in WT Th17-polarized cells (new Figure 7O). However, we also observed mild increase of ROR γ t and IL-17A expression in Th17-polarized cells from RORCE2-deficient and SOX5-BS mutant mice after Sox5 overexpression (new Figure 7O). These results support the notion that Sox5 has several potential target sites in Th17 cells. For instance, Sox5 together with c-Maf directly activates the promoter of ROR γ t in CD4⁺ T cells, which is very important for Th17 differentiation (J Exp Med. 2014 Aug 25; 211(9): 1857–1874). We have added the new results in new Figure 7O.

Minor concerns:

1. It should be cautious to call RORCE2 as an enhancer, as the authors did not have any evidences that RORCE2 is able to drive Rorc or irrelevant gene expression or promoter activities in a position-independent manner.

Response:

To address the reviewer's concern, we further investigated whether or not the enhancer candidates work in a position-independent manner in Th17-polarized cells. Specifically, we further cloned fragments containing RORCE1, RORCE2, RORCE3 and their combinations upstream of ROR γ t gene promoter or downstream of the luciferase gene in the reporter construct and detected the luciferase activity of these constructs in Th17-polarized cells. We observed the similar results in Th17 cells (new Figure 1F) with those results in EL4 cell lines. These results indicate that RORCE2 may act as a candidate enhancer for ROR γ t gene in a position-independent manner.

2. Figure 2B-G, Figure 7A-D, splenic CD4⁺ T cells are not ideal for detecting ROR γ t expression, due to the very low ROR γ t expression. Gut tissue would be much better.

Response:

Done. The results really showed the Th17 cell frequency was higher in CD4⁺ T cells of lamina propria in the small intestine than that in splenic CD4⁺ T cells. However, the trend of results in CD4⁺ T cells of lamina propria in the small intestine was consistent with that in the splenic CD4⁺ T cells. We have added the new results in revised Figure 2 and 7.

3. Figure 1C-D/9D, luciferase activity of RORCE2-construct should be shown in primary Th17 cells, if possible, as well as in Th1 and Th2 cells.

Response:

To address the reviewer's concern, we directly investigated the enhancer candidates in Th17-polarized cells. Specifically, we further cloned fragments containing RORCE1, RORCE2, RORCE3 and their combinations upstream of ROR γ t gene promoter or downstream of the luciferase gene in the reporter construct and detected the luciferase activity of these constructs in Th17-polarized cells. We observed the similar results in Th17 cells (new Figure 1F) with those results in EL4 cell lines. However, we don't further conduct the luciferase assay in primary Th1/2

cells because our ChIP-qPCR assays indicate RORCE2 is inactive in non-Th17 CD4⁺ T cells (Figure S2).

4. Figure 4C, the total average CD4⁺ T cells were ~30000 in WT mice, the RORγt⁺Th17 cells occupied ~10% total CD4⁺ T cells, it is unclear how to make an absolute number of ~20000 for CD4⁺RORγt⁺ Th17 cells in their assay.

Response:

We apologize for this mistake. It is really ~2000 of CD4⁺RORγt⁺ Th17 cells, not ~20000. We had miswritten the number previously. However, according to the reviewer's suggestion, we have replaced the CD4⁺RORγt⁺ phenotype of Th17 cells by CD4⁺IL-17A⁺ phenotype in the revised manuscript, and thus numbers for Th17 cells were changed accordingly (new Figure 4C). We thus replaced the previous CD4⁺ RORγt⁺Th17 results with CD4⁺IL-17A⁺Th17 results, the latter would reflect the Th17 function better because IL-17A is an effector of Th17 cells. Check please.

Reviewer #3 (Chromatin structure/architecture)(Remarks to the Author):

The identification of TH17-specific enhancers is a rather interesting topic due to intensive investigation of TH17-dependent pathologies. The authors, based on publically available data tried to identify the enhancer elements for the regulation of the ROR(γ)t transcription factor. They presented a plethora of research approaches including both biochemical in vitro as well as in vivo approaches. It is noteworthy that in support of their working hypotheses, the authors created and utilised many genetically modified mice or cell lines.

In certain parts of the manuscript there are overstatements that stem from the wrong interpretation of the data (such as PCR of genomic fragments that are deleted).

Moreover certain immunological assays seem not to be properly working (such as TH17 cell differentiation).

I have the following comments for the authors:

- Figure 1B: it is not defined what the authors mean by RORγt promoter. By what means is the gene's promoter defined?

Response:

We appreciate the reviewer's helpful comment. Actually, the RORγt promoter was defined in published paper (J Exp Med, 2011, 208:2321-2333), we have now cited this reference and described it in Methods section of the revised manuscript. So, in this study we didn't focus on the promoter definition further, but instead we focused on potential enhancer beyond the promoter region.

- Figure 1C/1D: It is an overstatement to talk about the significant effect of a potential enhancer when the proximal promoter transcriptional activity is increased by 2-3 fold maximum.

Response:

We are sorry for the overstatement and have deleted the word 'significant' in the context. Instead, we have now described the exact quantitative differences in the context throughout the manuscript.

-
- The statement of results part #1 regarding the identification of a TH17-specific enhancer is not supported by the data. Only transient transfection assays have been performed in EL4 and 293T cells (the cell types are not discussed at all, by the way) which cannot support the TH17-specific function of a potential regulatory element (that increases basic promoter activity by 3-fold) and assess its importance in TH17 cell differentiation and function.

Response:

It is well known that Th17 cells play key roles in many reported diseases, and ROR γ t is the signature transcription factor of Th17 cells. In this study, we intend to investigate how ROR γ t gene was regulated by cis-elements at the transcription level in Th17 cells. It has also been reported that ROR γ t gene is mainly expressed in Th17 cells, and certain ILCs types too. So, at the beginning of this study, we predicted the enhancer candidates in Th17 cells and ILCs by checking H3K4me2 modification (the enhancer predictor) upstream of ROR γ t promoter. As shown in Figure 1A, we observed the highly enriched H3K4me2 modification upstream of ROR γ t promoter in Th17 cells, but not in ILCs or Th1 cells. Therefore, we took the H3K4me2 enriched region as the enhancer candidate of ROR γ t gene and conducted subsequent verification experiments *in vitro* and *in vivo*. The transient transfection assays were just used to verify the enhancer potentiality of the selected candidate, which however was not aimed to verify the specificity of the candidate enhancer to Th17. However, the reviewer is exactly right that the candidate enhancer could not be termed as Th17-specific or ROR γ t-specific because this candidate enhancer might also interact with other target genes in Th17 cells or even in other unchecked cell types, although we could call it as “ROR γ t enhancer”. So, in the revised manuscript, we have changed the “Th17-specific enhancer” to “ROR γ t enhancer in Th17 cells”.

- In Figure 2C it is not clear how TH17 cells have been isolated for the intracellular staining of ROR γ t. For Fig2C/2D and others the authors do not refer to any quantitative differences in the results section but only refer to qualitative differences which they overestimate.

Response:

We thank the review for the helpful comments. In fact, we stained ROR γ t and IL-17A in the isolated CD4⁺T cells and determined the Th17 frequency by using CD4⁺ ROR γ t⁺ or IL-17A⁺ as the Th17 markers. So, we did not isolate Th17 cells for the intracellular staining of ROR γ t. The detailed information has been added in the revised M&M and the figure legend. Check please.

In addition, we have described the concrete quantitative difference in the results section in the revised manuscript.

- I assume that the authors in line 135 mean IL-4 in TH2 cells and IFN γ in TH1 cells.

Response:

The reviewer is exactly right. We have corrected the mistake in the revised manuscript.

- There is no information provided regarding the nature of utilized cell lines when first cited. For example, what is the origin of the B16 cell line which is presented completely different from the EL4 cell line.

Response:

We have now added more detailed information for all cell lines used in this study to avoid any confusion in the revised manuscript.

- In Figure 5, why do the authors used primers of opposite orientation of the anchor site and the test sites. Traditionally same orientation primers are utilized.

Response:

We apologize for the serious mistake. The reviewer is right and actually we used the same orientation primers of the anchor site and the test sites but we labeled them in the Figure 5 incorrectly. We have corrected the mistake in the revised version.

- Lines 169-170. The statement regarding the loss of loop scape structure due to the RORCE2 deletion is wrong. The loop is not lost, it is simply the DNA sequence deleted and therefore the PCR primers cannot anneal and amplify the respective regions. (The 3C PCR primers designed for the amplification of ROCE2 anneal in a DNA region from -5800 up to -4632. The knockout mouse bears a deletion from -5793 up to -4630).
- The same comment applies for the statement in lines 170-173 regarding the loss of loop formation in TH1 and TH2-polarised cells stemming from the KO mouse.
- The same comment applies for the statement in lines 186, 188-189 regarding the loss of loop formation in the ChIP-loop experiment with a SOX5 antibody in the RORCE2 deletion.

Response:

We appreciated for pertinent and helpful comments. After careful checking the issue of loss of loop upon RORCE2 deletion, we think the reviewer is exactly right. We really could not conclude that the loop loss was due to RORCE2 deletion at all, which is interpreted by the reviewer clearly. We apologize for our carelessness for this issue. So, we have deleted all the 3C-PCR results upon RORCE2 deletion, which actually did not make sense and deletion of these results would not affect our conclusion at all. We thank the reviewer again to help me correcting a mistake in this study.

- In supplemental Figure S5, it is not clear to me therefore I would like a comment from the authors regarding the high levels of Sox5 expression in EL4 cell without any stimulation (since these cells are not TH17 cells) while in T cells differentiated under neutral conditions Sox5 is not expressed (Fig.S5 left panel).

Response:

We are sorry not to describe the figure legend clearly. Actually, in Figure S5, B16 and EL4 cells had been stimulated with PI, but only EL4 expressed high level of Sox5 upon PI stimulation. We had not clearly described these conditions in the previous figure legend, which however have been now modified to describe more clearly in the revised figure legend. Check please.

- Figure 6C: please correct ROR γ t p (gamma).

Response: Done.

- In Figure 7 there seems to be an inherent problem of the whole FACS analysis. It is not clear whether the TH17 polarization conditions are good enough to provide cells of the TH17 cell lineage. The percentages of TH17 cell even in a wt genetic background are too low.

Response:

We thank the reviewer for the helpful comments on the FACS analysis and the Th17 polarization conditions.

As for low percentage of Th17 cells detected by FACS analysis in wild type mice spleens, we don't think it may be due to the inherent problem of FCM analysis. As indicated by another reviewer of this manuscript, it is probably that splenic CD4⁺ T cells are not ideal for detecting ROR γ t and IL-17A expression due to the very low ROR γ t expression, but gut tissue would be much better. So, we conducted the same experiments in mouse intestine, and we finally found that the Th17 cell frequency was much higher in CD4⁺ T cells of lamina propria in the small intestine tissue than that in splenic CD4⁺ T cells. However, the trend of results in CD4⁺ T cells of lamina propria in the small intestine was consistent with that in the splenic CD4⁺ T cells. We have added the new data in the revised Figure 7E-7H.

To address the reviewer's concerns for TH17 polarization conditions, we further optimized the protocol of *in vitro* differentiation of Th17 cells in the revised manuscript. Specifically, we have re-optimized the cell density of initial naïve CD4⁺ T cells, from 2×10⁶/ml to 1×10⁶/ml, according to the protocol of Th17 differentiation kit and the published literatures [J Exp Med. 2014 Aug 25;211(9):1857-74] because we found that 1×10⁶/ml of initial naïve CD4⁺T cells led to much less non-Th17 proliferation, compared with 2×10⁶/ml of initial naïve CD4⁺T cells. We further optimized the time of Th17 polarization from 5 days to 4 days, because the 5 days caused more cell death than 4 days in our experimental system; more importantly, 4 days are enough for the Th17 polarization as supported by the published literatures [J Immunol. 2014 Jan 1;192(1):110-22; Nat Commun. 2019 Nov 26;10(1):5450]. We ultimately found that there was more than 70% decrease for the frequency of Th17 in KO mice, in contrast to the previous results (less than 30% decrease) in KO mice. These results are shown in the new Figure 7I-7N, and should address the reviewer's related concerns.

Then we repeated all the related experiments basing on the above optimized conditions and modified all related results in the revised manuscript. Anyway, the trends of these new results were consistent with the previous version. We therefore thank the reviewer for the helpful comments on our study, which have strengthened the conclusion of our study.

- It is not clearly articulated what is the relationship of SOX5 and STAT3 proteins. Do they physically interact? Do they bind to nearby motifs synergistically?

Response:

Regarding to the relationship between Sox5 and STAT3, we conducted the Co-IP assay and found the interaction between the STAT3 and Sox5, suggesting that the Sox5 affected STAT3 recruitment by their interaction, considering that STAT3 binding site (STAT3-BS) in RORCE2 locates about 50bp downstream of the sox5 BS. The new data have been placed in Figure 9F.

REVIEWER COMMENTS

Reviewer #1 (Remarks to the Author):

The authors have performed an impressive number of experiments to address reviewers' concern. The results support the conclusions; however, the authors need to address the technical problem of Th17 cell differentiation and/or IL-17A intracellular cytokine staining before publication. It is not working and technical issues will undermine the conclusions of their study.

Here are some suggestions that may be helpful:

Th17 cell differentiation - try to use irradiated splenocytes (2000 rads) as feeder cells. If you do not have cell irradiator, you can block splenocyte proliferation using Mitomycin C, but expect that you will use 50% of cells.

Source of cytokine is also important. I provided this information for you as well.

1. Sort naive CD4+ T cells [CD4+ CD62L^{hi} CD25⁻]. We use CD25 staining because we want to remove Tregs from our naive CD4+ T cells, which are CD25⁺.
2. Activate sorted naive CD4 T cells with soluble anti-CD3 antibody (2 ug/ml; 145-2C11; BioXCell) in the presence of irradiated splenocytes (2000 rads) at a 5:1 ratio in a 24-well (1.5x10⁶ naive CD4+ T cells + 7.5x10⁶ irradiated splenocytes in 1 ml) and culture for 3 days in the presence of mIL-6 (20 ng/ml; Miltenyi Biotec), hTGFb1 (2 ng/ml; Miltenyi Biotec), anti-mouse IL-4 antibody (10 ug/ml; 11B11; BioXCell) and anti-IFN-g antibody (10 ug/ml; XMG1.2; BioXCell).
3. Feed and expand Th17 cell culture: on the third day of culture, transfer 4 24-wells into 1 10 cm-dish containing 10 ml of T cell media, mIL-6 (20 ng/ml; Miltenyi Biotec), hTGFb1 (2 ng/ml; Miltenyi Biotec), anti-mouse IL-4 antibody (10 ug/ml; 11B11; BioXCell) and anti-IFN-g antibody (10 ug/ml; XMG1.2; BioXCell). This time we supplement VERY LOW dose of hIL-2 (15 U/ml). We culture the cells for additional 2 days and perform intracellular cytokine staining. With this protocol, we obtained 50 - 60% IL-17A-producing cells with 90% viability.

IL-17A intracellular cytokine staining:

For analysis of cytokine production, cells are stimulated with PMA (phorbol 12-myristate 13 acetate; 50 ng/ml; Sigma) and ionomycin (1 uM; Calbiochem) for 4 hours total. Density of cells is 2x10⁶/ml.

The cells are stimulated with PMA+I for 2 hours WITHOUT cytokine blocker. Then, we add Monensin at a final concentration of 3 uM in the last two hours of stimulation. Mix gently. The intensity of IL-17A staining will be exceptionally bright. Most labs add monensin/brefeldin immediately when they stimulate cells fearing that cytokines will escape from the cells. It is a mistake. Monensin/brefeldin will cause cell stress and generally those cells do not produce a lot of cytokines. Using this protocol, where you stimulate cells without Golgi-stop allows for abundant cytokine production, then you block secretion for 2 hours. This could be your problem based on very low intensity IL-17A staining presented in dotplots. You tried to increase the signal by stretching the axes, but really, there is no need to do that if the staining works well.

We use anti-IL-17A antibody, sold by Thermo Fisher (formerly eBioscience). Clone # eBio17B7. You can choose PE at 1:200 dilution. We used Biolegend and BD Pharmingen anti-IL-17A antibodies, they

all work very well.

I hope that these recommendations improve the quality of your Th17 cell differentiation and/or IL-17A staining.

Reviewer #3 (Remarks to the Author):

The authors performed a great amount of experiments to address all three reviewers' comments. Moreover, there were at least 4 instances that mistakes have been made and have now been corrected by the authors after indicated by the reviewers.

I still have a few comments about the data presented.

1. Since the authors refer to the Transcription factor SOX-5 [UniProtKB - B2KFM9 (B2KFM9_MOUSE)] the designated nomenclature should be followed.
2. For the new Figure 4C the authors claim the effect of the RORCE2^{-/-} on TH17 differentiation. Though the total number of CD4⁺ cells is greatly affected in the knockout raising the possibility that the observed effect is due to the reduced number of CD4 cells and not a defect in TH17 differentiation or infiltration.
3. Comparing new Figures 4D to 4E one can see that in 4E the RORCE2^{-/-} has fewer CD4⁺ cells which is completely different from Figure 4D (about the same number of CD4⁺ cells between wt and ko mice). Why the number of cells analyzed by FACS is different between the two genotypes?
4. Figure 9F. Why the coIP experiment was performed in a cell line instead of the relevant primary cell type?

Reviewer #1 (Remarks to the Author):

The authors have performed an impressive number of experiments to address reviewers' concern. The results support the conclusions; however, the authors need to address the technical problem of Th17 cell differentiation and/or IL-17A intracellular cytokine staining before publication. It is not working and technical issues will undermine the conclusions of their study.

Here are some suggestions that may be helpful:

Th17 cell differentiation - try to use irradiated splenocytes (2000 rads) as feeder cells. If you do not have cell irradiator, you can block splenocyte proliferation using Mitomycin C, but expect that you will use 50% of cells.

Source of cytokine is also important. I provided this information for you as well.

1. Sort naive CD4⁺ T cells [CD4⁺ CD62L^{hi} CD25⁻]. We use CD25 staining because we want to remove Tregs from our naive CD4⁺ T cells, which are CD25⁺.
2. Activate sorted naive CD4 T cells with soluble anti-CD3 antibody (2 ug/ml; 145-2C11; BioXCell) in the presence of irradiated splenocytes (2000 rads) at a 5:1 ratio in a 24-well (1.5x10⁶ naive CD4⁺ T cells + 7.5x10⁶ irradiated splenocytes in 1 ml) and culture for 3 days in the presence of mIL-6 (20 ng/ml; Miltenyi Biotec), hTGFb1 (2 ng/ml; Miltenyi Biotec), anti-mouse IL-4 antibody (10 ug/ml; 11B11; BioXCell) and anti-IFN-g antibody (10 ug/ml; XMG1.2; BioXCell).
3. Feed and expand Th17 cell culture: on the third day of culture, transfer 4 24-wells into 1 10 cm-dish containing 10 ml of T cell media, mIL-6 (20 ng/ml; Miltenyi Biotec), hTGFb1 (2 ng/ml; Miltenyi Biotec), anti-mouse IL-4 antibody (10 ug/ml; 11B11; BioXCell) and anti-IFN-g antibody (10 ug/ml; XMG1.2; BioXCell). This time we supplement VERY LOW dose of hIL-2 (15 U/ml). We culture the cells for additional 2 days and perform intracellular cytokine staining. With this protocol, we obtained 50 - 60% IL-17A-producing cells with 90% viability.

IL-17A intracellular cytokine staining:

For analysis of cytokine production, cells are stimulated with PMA (phorbol 12-myristate 13 acetate; 50 ng/ml; Sigma) and ionomycin (1 uM; Calbiochem) for 4 hours total. Density of cells is 2x10⁶/ml.

The cells are stimulated with PMA+I for 2 hours WITHOUT cytokine blocker. Then, we add Monensin at a final concentration of 3 uM in the last two hours of stimulation. Mix gently. The intensity of IL-17A staining will be exceptionally bright. Most labs add monensin/brefeldin immediately when they stimulate cells fearing that cytokines will escape from the cells. It is a mistake. Monensin/brefeldin will cause cell stress and generally those cells do not produce a lot of cytokines. Using this protocol, where you stimulate cells without Golgi-stop allows for abundant cytokine production, then you block secretion for 2 hours. This could be your problem based on very low intensity IL-17A staining presented in dotplots. You tried to increase the signal by stretching the axes, but really, there is no need to do that if the staining works well.

We use anti-IL-17A antibody, sold by Thermo Fisher (formerly eBioscience). Clone # eBio17B7. You can choose PE at 1:200 dilution. We used Biolegend and BD Pharmingen anti-IL-17A antibodies, they all work very well.

I hope that these recommendations improve the quality of your Th17 cell differentiation and/or IL-17A staining.

Response:

We appreciate the reviewer's kind and helpful suggestions about the detailed protocols for Th17 cell differentiation and IL-17A staining. In the revised work, we have followed the methods suggested by the reviewer and found that the quality of Th17 cell differentiation and IL-17A staining was indeed greatly improved, especially for the intensity of IL-17A staining (new Figure 3A-3E and Figure 7I-7K). We thank the reviewer for the wonderful suggestions, which really further support the conclusion in this study.

Figure 3

Figure 3. RORCE2 deficiency inhibited Th17 cell polarization *in vitro*.

A. Naïve splenic CD4⁺ T cells from RORCE2^{-/-} or WT mice were stimulated under Th17-polarizing conditions for 5 days. The frequency of CD4⁺RORγt⁺ Th17 cells was measured by flow cytometry. B and C. Summaries for the percentages of CD4⁺RORγt⁺ Th17 cells (B) and the MFI of RORγt in Th17 cells (C) generated from RORCE2^{-/-} and WT mice on day 5 after *in vitro* polarization are shown. D. Naïve splenic CD4⁺ T cells from RORCE2^{-/-} or WT mice were stimulated under Th17-polarizing

conditions for 5 days. The frequency of CD4⁺IL-17A⁺ Th17 cells was measured by flow cytometry. E. Summaries for the percentages of CD4⁺IL-17A⁺ Th17 cells generated from RORCE2^{-/-} and WT mice on day 5 after *in vitro* polarization are shown. Error bars show the mean \pm SD. *, $P \leq 0.05$; **, $P \leq 0.01$; ***, $P \leq 0.001$; n.s., not significant. Student's *t* test. Source data are provided as a Source Data file.

Figure 7

Figure 7. SOX-5-BS deletion in RORCE resulted in reductions in RORγt expression and Th17 cell differentiation.

I-K. Splenic naïve CD4⁺ T cells from SOX-5-BS^{-/-} or WT mice stimulated under Th17 polarizing conditions for 4 d. The frequency of CD4⁺RORγt⁺ Th17 (I) and CD4⁺IL-17A⁺ Th17 (J and K) cells and MFI of RORγt (I) were measured by flow cytometry and summarized.

Error bars show the mean \pm SD. *, $P \leq 0.05$; **, $P \leq 0.01$; ***, $P \leq 0.001$; n.s., not significant. Student's *t* test. Source data are provided as a Source Data file.

Reviewer #3 (Remarks to the Author):

The authors performed a great amount of experiments to address all three reviewers' comments. Moreover, there were at least 4 instances that mistakes have been made and have now been corrected by the authors after indicated by the reviewers.

I still have a few comments about the data presented.

1. Since the authors refer to the Transcription factor SOX-5 [UniProtKB - B2KFM9 (B2KFM9_MOUSE)] the designated nomenclature should be followed.

Response:

According to the suggestion, we have carefully corrected the designated nomenclature throughout the manuscript.

2. For the new Figure 4C the authors claim the effect of the RORCE2^{-/-} on TH17 differentiation. Though the total number of CD4⁺ cells is greatly affected in the knockout raising the possibility that the observed effect is due to the reduced number of CD4 cells and not a defect in TH17 differentiation or infiltration.

Response:

We appreciate the reviewer's helpful comments. The CD4⁺T cells consist of several subsets such as Th1, Th2 and Th17 cells. However, Th17 cells have been verified to play the pivotal role in EAE pathology. Cua and colleagues found that mice lacking the p19 chain of IL-23, required for development of Th17 cell responses, but not the p35 chain of IL-12, required for development of Th1 cells, were resistant to the induction of EAE (Nature, 2003, 421:744). Furthermore, auto-antigen-specific Th17 cells induced EAE following transfer into naïve mice (Nat Rev Immunol, 2011, 11:807-822). These studies support the unique role of Th17 cells in EAE and pave the way for a series of discoveries on the key pathogenic role of IL-17 and Th17 cells in many autoimmune diseases. All these studies suggest the frequency and cell number of Th17 cells might be a potential indicators of EAE pathology (Nat Commun, 2016; 7: 12993; J Exp Med, 2014, 211:1857; J Exp Med, 2017, 214:1453). Therefore, in this study we mainly analyze the changes of Th17 cells in the prototype EAE mice model.

However, to exactly address the reviewer's concern, we conducted further experiments to confirm the unique IL-17-related effects on EAE in this study by referring to the published paper (Nat Commun, 2016; 7: 12993), which demonstrates the similar data for Th17 and CD4⁺T cells in the same EAE model with the present study. We immunized RORCE2^{-/-} and WT mice with MOG₃₅₋₅₅ peptide and extracted draining lymph nodes on day 8. Isolated cells from the lymph nodes were further cultured in *ex vivo* with MOG for 3 days. IL-17A, IFN- γ and IL-4 concentration were measured by ELISA. As shown in new Figure 4F, IL-17A production was decreased in RORCE2-deficient mice, whereas IFN- γ and IL-4 production was unaffected. These new data further support the significance of RORCE2 in the generation of Th17 cells and the production of IL-17, rather than other CD4⁺ T cell subsets such as Th1/Th2 cells. We hope the above explanation and the new experimental data have sufficiently addressed the reviewer's concerns.

Figure 4

F

Figure 4. RORCE2 deficiency ameliorated neuroinflammation in EAE mice.

F. Mononuclear cells were collected at day 8 from inguinal lymph nodes and further cultured *ex vivo* with MOG for 3 days. The concentration of IL-17A, IFN γ and IL-4 was measured by ELISA, respectively. Error bars show the mean \pm SD. **, P < 0.01; n.s., not significant. n = 5 in each group; Student's t test. Source data are provided as a Source Data file.

3. Comparing new Figures 4D to 4E one can see that in 4E the RORCE2^{-/-} has fewer CD4⁺ cells which is completely different from Figure 4D (about the same number of CD4⁺ cells between wt and ko mice). Why the number of cells analyzed by FACS is different between the two genotypes?

Response:

We thank the review for the cell number issue in Figure 4E. Actually, what we focused on in Figures 4D and 4E was the frequency of the subsets of CD4⁺T cells, and thus unsuitably ignored the input cell number for the FCM analyses between the WT and KO mice. To address the reviewer's concerns, we have further performed the FCM analyses in both groups by using the same input number of cells. Expectably, the new FCM data are very similar with those in previous manuscript. All the FCM figures in Figure 4 and Figure 8 have been replaced (new Figure 4C-4E and Figure 8C-8E)

Figure 4

Figure 4. RORCE2 deficiency ameliorated neuroinflammation in EAE mice.

C. On day 30 after EAE induction, the CD4⁺ T cells among the leukocytes isolated from the spinal cord of

the indicated mice were gated and further analyzed to determine the frequencies of CD4⁺IL-17A⁺ Th17 cells. In addition, the absolute numbers of spinal cord-infiltrated CD4⁺IL-17A⁺ Th17 cells and total CD4⁺ T cells were also evaluated by flow cytometry.

D-E. On day 30 after EAE induction, the CD4⁺ T cells among the leukocytes isolated from the spinal cord of the indicated mice were gated and further analyzed to determine the frequencies of CD4⁺IFN γ ⁺ Th1 and CD4⁺IL-4⁺ Th2 cells by flow cytometry.

Error bars show the mean \pm SD. **, $P \leq 0.01$; n.s., not significant. n = 5 in each group; Student's t test.

Source data are provided as a Source Data file.

Figure 8

Figure 8. SOX-5-BS deletion reduced the severity of EAE

C and D. On day 30 after EAE induction, the CD4⁺ T cells among the leukocytes isolated from the spinal cord of the indicated mice were gated and further analyzed to determine the frequencies of CD4⁺IL-17A⁺ Th17 cells.

E. The absolute numbers of spinal cord-infiltrated CD4⁺IL-17A⁺ Th17 and total CD4⁺ T cells were evaluated by flow cytometry.

Error bars show the mean \pm SD. **, $P \leq 0.01$; n.s., not significant. n = 5 in each group; Student's t test.

Source data are provided as a Source Data file.

4. Figure 9F. Why the coIP experiment was performed in a cell line instead of the relevant primary cell type?

Response:

In our previous manuscript, the co-IP experiment was conducted in Hela cells because it is easier to conduct such experiment in a cell line than the primary Th17 cells, for instance, the Th17 cells are relatively hard to be obtained than the transferred Hela cells. However, to address the reviewer's concern, we further conducted the co-IP experiments in primary Th17 cells that were differentiated from naïve CD4⁺T cells under Th17-polarization condition. As expected, we got the similar results in Th17 cells with those in Hela cells. The new results have been added in new **Figure 9G** of the revised manuscript.

Figure 9

Figure 9. STAT3 was necessary for the activation of RORCE2.

G. Whole-cell lysates from Th17 cells were subjected to IP with anti-STAT3 antibody or control rabbit IgG and IB with anti-SOX-5 or anti-STAT3 antibody. Input proteins (input) were also IB with anti-SOX-5 or anti-STAT3 antibody. Data are representative of three experiments. Source data are provided as a Source Data file.

REVIEWERS' COMMENTS

Reviewer #1 (Remarks to the Author):

The authors have addressed my main concern regarding IL-17A staining.

Reviewer #3 (Remarks to the Author):

The authors have satisfactorily addressed all the concerns raised by the three reviewers.

REVIEWERS' COMMENTS

Reviewer #1 (Remarks to the Author):

The authors have addressed my main concern regarding IL-17A staining.

Response:

We appreciate the reviewer's positive comments for our manuscript. Thanks again.

Reviewer #3 (Remarks to the Author):

The authors have satisfactorily addressed all the concerns raised by the three reviewers.

Response:

We appreciate the reviewer's helpful comments. Thanks a lot.